# Are There Barriers Separating the Pink River Dolphin Populations (*Inia boliviensis*, Iniidae, Cetacea) within the Mamoré–Iténez River Basins (Bolivia)? An Analysis of Its Genetic Structure by Means of Mitochondrial and Nuclear DNA Markers

**DOI:** 10.3390/genes15081012

**Published:** 2024-08-01

**Authors:** Manuel Ruiz-García, Pablo Escobar-Armel, María Martínez-Agüero, Magda Gaviria, Diana Álvarez, Myreya Pinedo, Joseph Mark Shostell

**Affiliations:** 1Laboratorio de Genética de Poblaciones Molecular-Biología Evolutiva, Unidad de Genética, Departamento de Biología, Facultad de Ciencias, Pontificia Universidad Javeriana, Carrera 7A No 43-82, Bogotá 110311, DC, Colombia; p.escobar@javeriana.edu.co (P.E.-A.); mgaviria@javeriana.edu.co (M.G.); dalvarez@javeriana.edu.co (D.Á.); mozpinedo@gmail.com (M.P.); 2Facultad de Ciencias Naturales y Matemáticas, Universidad del Rosario, Bogotá 111321, DC, Colombia; maria.martinez@urosario.edu.co; 3Math, Science and Technology Department, University of Minnesota Crookston, Crookston, MN 56716, USA; shostell@crk.umn.edu

**Keywords:** *DQB-1* gene sequences, *Inia boliviensis*, microsatellites, mitochondrial control region, Pleistocene, rapids, South American climatic and geological changes, spatial patterns

## Abstract

The pink river dolphin, or bufeo, is one of the dolphins which lives in the rivers of the Orinoco and Amazon basins in South America. The Bolivian bufeo population is considered a differentiated species (*Inia boliviensis*) from the Amazon and Orinoco species (*Inia geoffrensis*). Until now, no study has completed an extensive population genetics analysis of the bufeo in Bolivian rivers. We analyzed 82 bufeos from different rivers from the Mamoré and Iténez (Guaporé) river basins for the mt control region (CR), nuclear microsatellites, and *DQB-1* gene sequences to determine if the inner rapids of these Bolivian river basins have some influence on the genetic structure of this species. The first relevant result was that the genetic diversity for CR, and the microsatellites were substantially lower in the *Bolivian bufeos* than in the dolphins studied in other areas of the Amazon and Orinoco. However, the *DQB-1* gene sequences yielded similar genetic diversity to those found in other areas. The second relevant result is the existence of some significant genetic heterogeneity among the bufeo populations within Bolivia, although in a small degree, but this differentiation is independent of the inner rapids of the Bolivian rivers we sampled. The third relevant result was the existence of significant isolation by distance for the CR, but not for microsatellites and *DQB-1* gene sequences. This was related to differential gene flow capacity of females (philopatric) and males (less philopatric and more migrants) and, possibly, to different selective patterns affecting the molecular markers studied. The fourth relevant result was related to diverse demographic changes of these bufeos. At least two or three bottleneck events and one or two population expansions have occurred in the Bolivian bufeo population. The major part of these events occurred during the Pleistocene.

## 1. Introduction

The pink river dolphin, Amazon River dolphin, bufeo, or boto, is a species (*I. geoffrensis*) widely distributed in the rivers of the Amazon and Orinoco basins [1,2,3,4,5]. The range of distribution in South America includes rivers in Bolivia, Brazil, Colombia, Ecuador, Peru, Venezuela, and French Guyana, covering an area of approximately 7 million km^2^ [6]. In both basins, bufeos are found from the headwaters to the delta [2]. 

Traditionally, it was thought that the main limits to the distribution of bufeos were large rapids and waterfalls [7]. In accordance with this idea, Banguera-Hinestroza et al. (2002) [3] was the first group of researchers to demonstrate, with a relatively large sample, that the mitochondrial (mt) DNA haplotypes of the Bolivian *Inia* were reciprocally monophyletic with respect to both the Orinoco and Amazon *Inia*. More recently, the mtDNA monophyly of the Bolivian bufeos has been ratified [4,8,9]. They also used nuclear intron sequences and nuclear microsatellite markers to confirm the reciprocal monophyletic relationship between the Bolivian bufeos and the Amazon and Orinoco bufeos. These results were considered when the decision was made to elevate the Bolivian form to the status of full species (*I. boliviensis*). Therefore, *I. boliviensis* has a much more restricted distribution than *I. geoffrensis* and is only found in the rivers of northern and northeastern Bolivia, including the Abuná, Beni, Iténez-Guaporé and Mamoré River basins that form a political boundary between Bolivia and Brazil [3,4,9]. However, the results of new studies of the Bolivian bufeo [9,10,11] and this study suggest more the status of a subspecies (*I. geoffrensis boliviensis*), rather than a full species. However, we will still employ the denomination of *I. boliviensis* throughout this text.

On the other hand, river dolphins are among the most threatened cetaceans on the planet [12,13]. They inhabit geographical areas of South America and Asia that have undergone intensive transformations due to the impact of human activities in the last century [14,15]. In fact, in 2005, the Chinese river dolphin, the baiji, (*Lipotes vexillifer*) became functionally extinct [16,17,18] in the Yangtze River. Two other species of Asian dolphins are at risk of extinction as well. One is *Platanista gangetica* (endangered), which is distributed across Bangladesh (Brahmaputra and Megna River systems). The second is *Platanista minor* (endangered), from the Indo River basin (Pakistan) [19]. A third species, the Irrawaddy dolphin (*Orcaella brevirostris*) is currently classified as vulnerable. Additionally, the bufeos are considered endangered [20] and are among the most threatened aquatic mammals. Several anthropogenic activities threaten the bufeos: (1) one of the most serious threats is accidental capture by the fishing industry [21]; (2) dolphins are also intentionally captured and killed as sources of “love charms”. We have observed the sexual organs (especially from females) of bufeos for sale in the “Pasaje Paquito” of the Belem market, as well as in the “artesanal San Juan” at Iquitos (Perú); (3) bufeos are also killed and used as bait to catch catfish species like the “mota” or “mapurito” (*Calophysus macropterus*) in the Orinoco and Amazon Rivers (mainly in Colombia and Brazil) [20,22]; (4) ecological destruction and habitat fragmentation by deforestation are serious threats to bufeos. For example, the deforestation process in the Amazon causes an annual loss rate of 0.4–2.3% [23]. Deforestation affects the hydrological cycle. In fact, destruction of the forests affects the ability of the Amazon to recycle 48% of the precipitation it receives [24]; (5) mercury is a threat to bufeos too [25]; (6) oil spills are threats to bufeos too, because these, along with gas and fuel oil explorations, have undoubtedly impacted the food chains. For instance, it was revealed that there was a population density declination of *Inia* in the Cuyabeno and Lagartococha Rivers of the northeastern Ecuadorian Amazon from 1996 to 1998. This declination seemed to be due to contamination by six oil spills and waste waters of the oil fields [26]; (7) another extreme threat to river dolphins is dam construction. Dams wholly transform the hydrologic cycle of a river, thereby altering flood patterns (occurrence and size) and affecting precipitation patterns. Dams act as barriers to fish migrations. They also affect sediment load and water quality of rivers and suppress the seasonal flow peaks, preventing the formation of adjacent floodplains which are extremely vital for the subsistence of bufeos in certain years and seasons [27]. Dams also constitute absolute barriers to the potential movements of bufeos [28]. Two dams exist within the distribution range of the *Bolivian bufeos* in the Madeira River (Brazil) [10,11]; (8) fishing is a threat to the bufeos. Nylon gill nets catch more fish with the fishermen expending less effort. Small nylon-mesh gill nets are extremely damaging because they indiscriminately catch all sizes and age-classes of fish. Such use of nets can have major and negative effects on river dolphin prey. The effects of fishing on bufeos are compounded due to the congregation of bufeos in confluence areas [29], where nets are often deployed. Dolphins are vulnerable to accidental entanglement in fishing nets (especially with nylon gill nets) [29]; (9) the capture of specimens for export to aquaria has historically been a problem, but less so today. Since 1956, over 100 bufeos have been captured and shipped to aquaria [30,31]. Shipping and imports of bufeos have ceased, due to expensive costs, high mortality, and new conservation laws; (10) collision with boats is another threat for bufeos; and (11) climate change and the encroachment of human settlements are additional threats to bufeos [32].

In the current work, we collected tissue samples from 82 bufeos inhabiting rivers of the Mamoré and Iténez (Guaporé) River basins. The tissues were analyzed for different kinds of molecular markers [mt control region (CR), nuclear microsatellites, and *DQB-1* gene sequences]. The objective of this analysis was to determine if spatial structure has any influence on the genetic structure of this species in these Bolivian rivers. If it does have influence, we suggest two explanations. The first is the existence of many rapids and waterfalls which can act as obstacles to gene flow among populations of *Bolivian bufeos*. These barriers have an important impact on the movement of fishes [33,34,35], turtles and caimans [36,37], the Amazonian manatee [38], and the tucuxi [39]. The second is the isolation of genetic patches due to distance. In these cases, genetic patches are caused by biological events rather than physical barriers such as rapids or waterfalls. In fact, despite an apparent lack of physical barriers to dispersal, many aquatic organisms exhibit population structure over scales smaller than their dispersal potential [40]. For instance, some species of marine turtles [41] and cetaceans [42,43,44] showed this pattern. This could be caused by differential adaptation to the environmental complexity of certain aquatic systems [45,46,47], isolation-by- distance [48], or strong regional demographic variations [49]. Indeed, it was asserted that, in some seas (the Mediterranean, Eastern North Atlantic, and the North Sea), the population structure of certain species of dolphins appears to correlate strongly with environmental differences, which agrees well with suggestions that differences in habitat requirements drive population genetic structure in some cetaceans [42].

Taking into account all of these facts, the main aims of the current study are as follows: (1) to estimate the levels of genetic diversity of the bufeos throughout Bolivian rivers using different kinds of molecular markers (CR, nuclear microsatellites, and *DQB-1* gene sequences); (2) to estimate the genetic heterogeneity among different populations of bufeos throughout Bolivian rivers using different kinds of molecular markers; (3) to determine if gene flow is considerably higher than genetic drift among populations of bufeos throughout Bolivian rivers using nuclear microsatellites; (4) to determine how many different significant populations or “gene pools” of *I. boliviensis* are detectable throughout Bolivian rivers using different kinds of molecular markers; (5) to find possible significant spatial genetic structure among different populations of bufeos throughout Bolivian rivers using different kinds of molecular markers and to determine if this spatial genetic structure is related to the presence of rapids and waterfalls in the Bolivian rivers; and (6) to determine possible demographic changes over time for *I. boliviensis* throughout Bolivian rivers using different kinds of molecular markers and to correlate these demographic changes with past geological, climatological, and hydrological changes.

## 2. Materials and Methods

### 2.1. Studied Area and Samples

The Mamoré sub-basin (611,800 km^2^) covers 66.7% of the Upper Madeira River Basin and includes 47.7% of its volume and has a length of 1930 km [50]. The Mamoré River is a typical whitewater river, containing a considerable amount of small particles and inorganic suspended solids that limit water visibility. The hydrological pattern is closely related to precipitation, showing a unimodal curve with a high-water period between December and April and a low-water period from June to October. Fluctuations in water level of up to 10 m result in flooding of the forest or surrounding area [51].

We sampled 82 bufeos in different rivers of the Mamoré sub-basin (Figure 1). In the main Mamoré River, 54 bufeos were sampled in four different areas. We named these areas as “populations” for operative purposes, but they are really sampling sites. However, these sampling sites belonged to very precise geographical areas, to different rivers, or were separated by different rapids. In fact, it is not very important to define a priori operating populations because later analyses (such as phylogenetic trees, AMOVA or BAPS) will determine exactly how many different populations there are. One was at the confluence of the Ibaré and Mamoré Rivers (latitude: −14.550, longitude: −65.003; 36 specimens). A second was in the middle of the Mamoré (−13.445, −65.235) and included four sampling points that were close together that were not separated by geographical barriers. These were Porvenir, Bellaunión, Cerrito, and the confluence of Iruyañez River with the Mamoré River. Ten specimens were collected. The third was in Bolivar (−12.393, −65.159; 3 specimens). The fourth was El Corte (−11.925, −65.054; 5 specimens). In addition, some bufeos were sampled in three affluent tributaries of the Mamoré River. They were the Securé River (−15.333, −65.020; 3 specimens), the upper Ibaré River (−14.333, −64.900; 5 specimens), and the Tijamuchí River (−14.450, −65.900; 2 specimens). In the main Iténez (Guaporé) River, 2 specimens were sampled at a place named El Azul (−11.972, −65.036). Bufeos were also sampled in two affluent tributaries of the Iténez (Guaporé) River—the San Martín River (−13.300, −63.417; 2 specimens) and the Ipurupuru River (−14.302, −65.050; 14 specimens). An important fact is that the Middle Mamoré is separated from Bolivar by two rapids (Cachuela Matucaré and Cachuela Warnes). Similarly, Bolivar is separated from El Corte by two rapids (Cachuela Mayo—Mayo and Cachuela Envidia). El Corte is separated from El Azul by one rapid (Cachuela San José). Thus, we want to know if these rapids influence the genetic structure of *I. boliviensis* in the Bolivian rivers.

The capture, transporting, and ethical permissions to obtain samples of the bufeos were given by the Ministerio de Desarrollo Sostenible y Planificación, Dirección General de Biodiversidad from Bolivia (DGB/UVS No 477/03; approval date 27 May 2003) and CITES Bolivia (B09118259), and Ministerio de Producción (Lima, Peru) (No 402-2003-PRODUCE/DNEPP; approval date 13 November 2003).

The bufeos were captured using special fishing nets (80 m × 5 m), taking special care to ensure the physical safety of each bufeo. Three indigenous Bolivian fishermen and five biologists worked together in the water to capture each bufeo. The individuals were brought on board (in a little wood “canoa”) and their caudal fins were biopsied. After the biopsy, the wound was covered with an antibiotic. Each specimen was measured for different biometric characteristics and safely released within 5–8 min of capture. The specimens were marked to avoid any recapture. The biopsies were stored in absolute alcohol until DNA extraction. This task (to capture the specimens) was hard, difficult, and dangerous, because the Bolivian rivers were infected with piranhas, caimans, and anacondas and the indigenous fishermen and the biologists were forced to live in the unpopulated Bolivian jungle for three months. All this did not allow us to obtain the same number of samples from each of the populations analyzed, since, in some places, there were more bufeos and it was, depending on the conditions of the river, easier to capture them. However, in other places, the bufeos were scarcer and the conditions of the river were much more complex to capture them. When it was possible, we employed unbiased statistics for sample sizes.

### 2.2. Molecular Markers

DNA extraction from the caudal fin biopsies was performed by the phenol–chloroform method [52].

#### 2.2.1. Mitochondrial Control Region

For the analysis of the CR (400 base pairs, bp, were sequenced), 82 specimens were used. The primers used were H16498 and TRO [3]. The sequences of these primers were 5′ CCT GAA GTA AGA ACC AGA TG3′ for H16498 and 5′ CCT CCC TAA GAC TCA AGG 3′ for TRO. The PCR reactions were undertaken in a final 25 μL volume with the following conditions: 4 pmol of each primer, 2.5 μL of reaction buffer (1×), 3.0 μL of 2.5 mM MgCl_2_, 0.25 mM of each dNTP, one unit of Gold Taq Polymerase (Promega), and 2 μL of (25–50 ng per μL) of DNA. The amplifications were carried out in a BioRad thermocycler with the following protocol: 95 °C for 5 min, 30 cycles at 95 °C for 45 s, at 52 °C for 45 s and at 72 °C for 45 s, and at 72 °C for 10 min for final extension. All DNA fragments were sequenced in both directions, and in cases of mismatching the PCR reaction was repeated, and the product sequenced again.

#### 2.2.2. Microsatellites

Ten microsatellite markers were studied for 61 *Bolivian bufeos* sampled: *Ev14*, *Ev37*, *Ev76*, *Ev94,* and *Ev96* [53], *MK5* [54], *PPHO137* [55], *KWM2a*, *KWM2b* and *KWM12a* [56,57]. The reactions were completed in 25 μL with the following conditions: 10 pmol of forward and reverse primers, 2.5 μL of reaction buffer (10×), 3.0 μL of MgCl_2_ 3 mM, 1 μL of dNTPs 1 μM, one Taq polymerase unit, 13.5 μL of H_2_O, and 2 μL between 25 and 50 ng per μL of DNA. The PCR conditions were 95 °C for 5 min, a number of determined cycles of 1 min at 95 °C (see the next sentence), 1 min at the most accurate annealing temperature (see in the next sentence), 1 min at 72 °C, and 5 min at 72 °C. The number of cycles and the annealing temperatures were as follows: *Ev14* (35 cycles at 58 °C), *Ev37* (10 cycles at 46 °C and 25 cycles at 56 °C), *Ev76* (10 cycles at 46 °C and 25 cycles at 56 °C), *Ev94* (10 cycles at 46 °C and 25 cycles at 56 °C), *Ev96* (10 cycles at 46 °C and 25 cycles at 56 °C), *MK5* (35 cycles at 56 °C), *PPHO137* (35 cycles at 57 °C), *KWM2a* (35 cycles at 57 °C), *KWM2b* (35 cycles at 54 °C) and *KWM12a* (35 cycles at 55 °C). The PCR amplification products were run in denaturant 6% polyacrylamide gels within a Hoefer SQ3 sequencer vertical chamber. Gels migrated for 2–3 h depending on marker size and were then stained with AgNO_3_ (silver nitrate). Every sixth line in the gel contained molecular markers (ϕ174 cut with Hind III and Hinf I). The PCR amplifications were performed in triplicate to ensure the accuracy of the genotypes obtained.

#### 2.2.3. DQB-1 Gene Sequences

We amplified a 172 bp fragment of the variable exon 2 of *DQB-1* gene via PCR, using the previously published primers and conditions [58] for 23 *Bolivian bufeos*. PCR products of the expected size were discriminated using the SSCP (single-strand conformational polymorphism) method [58]. After electrophoresis, the gel was stained in silver and the patterns were characterized. All PCR products displaying unique SSCP patterns were cloned (pCR2,1 vector; Topo TA cloning kit, Invitrogen, Waltham, MA, USA) and Sanger-sequenced on both strands. The histocompatibility of the sequences was confirmed based on their shared homology with other cetacean *DQB-1* gene sequences available on GenBank, using NCBI Blast tool.

### 2.3. Mathematical Population Analyses

#### 2.3.1. Genetic Diversity

Genetic diversity statistics were applied to CR, and the sequences at the *DQB-1* locus, to determine the genetic diversity for the overall sample of *I. boliviensis* studied: number of haplotypes (H), haplotype diversity (H_d_), nucleotide diversity (π), average number of nucleotide differences (k) and θ statistic by sequence. These genetic diversity statistics were calculated with the DNAsp 5.1 [59] and Arlequin 3.5.1.2 programs [60].

For microsatellites, we estimated the mean number of alleles per locus (MNA), and the observed heterozygosity (H_o_) and the expected heterozygosity (H_e_) [61] were calculated for the overall sample studied. H_e_ values were arcsine transformed prior to statistical analysis [62]. Kruskal–Wallis (KW) tests were performed to determine significant differences in genetic diversity among populations [63]. If significant differences were found, a Dunn’s test was performed on pairwise comparisons to determine which pair of groupings was driving the significant difference [63,64].

#### 2.3.2. Hardy–Weinberg Equilibrium

The Hardy–Weinberg equilibrium (HWE) for the *I. boliviensis* populations was estimated using several different strategies for microsatellites and the *DQB-1* locus. Robertson and Hill (1984)’s f statistic (R-H) was used to calculate the degree of excess or deficit of homozygosity and heterozygosity (complete enumeration) of the overall sample [65]. To measure the exact probabilities of the G test, the Markov chain method, with a 10,000-dememorization number, 20 batches and 5000 iterations per batch, was used, following the Genepop v4.2.1 Program [66]. The HWE was simultaneously analyzed by locus and by population using Fisher’s method [66].

#### 2.3.3. Phylogenetic Trees for Mt Control Region Sequences

The jModelTest v2.0 [67], Kakusan4 [68], and MEGA X v10.0.5 [69] were applied to determine the best evolutionary substitution nucleotide model for the sequences analyzed for CR. BIC [70] and AIC [71] were used to determine the best evolutionary nucleotide model. A phylogenetic tree was constructed using maximum likelihood (ML) procedure. The ML tree was obtained using RAxML v8.2.X [72] implemented in the CIPRES Science Gateway [73]. The GTR ± G model was used to search the ML tree. We estimated node support using the rapid-bootstrapping algorithm (−fa-x option) for 1000 non-parametric bootstrap replicates [74]. The haplogroups of *Bolivian bufeos* were considered significant when bootstraps were higher than 80% (lax limit) [75].

#### 2.3.4. Genetic Heterogeneity

For CR, we estimated the following statistical heterogeneity indices: table of contingency, H_ST_, K_ST_, K_ST*_, γ_ST_, N_ST_, and F_ST_ [76] for the overall sample of Bolivian bufeos studied. Indirect gene-flow estimates were obtained assuming an infinite island model [77]. Significance was estimated with permutation tests using 10,000 replicates. We also estimated genetic heterogeneity between bufeo population pairs. For this task, we used exact probability tests with Markov chains, using 10,000 dememorization parameters, 20 batches, and 5000 iterations per batch. All the heterogeneity statistics were calculated with the DNAsp v5.1 [59] and Arlequin v3.5.1.2 programs [60]. We also used the Kimura 2P genetic distance [78] to determine the percentage of genetic differences among the different *I. boliviensis* populations studied. The Kimura 2P genetic distance is a standard measurement for barcoding tasks [79,80]. AMOVAs (analyses of molecular variance) were used as another procedure to measure the genetic heterogeneity among the different populations of *I. boliviensis*. The AMOVAs allowed us to investigate population subdivision [81], using k (number of possible populations) from 2 to 7.

For microsatellites, the first procedure to detect some possible genetic heterogeneity among the Bolivian bufeo populations was a Correspondence Factorial Analysis (CFA) with the GENETIX 4.05 [82]. Additionally, a second procedure to determine genetic heterogeneity among the *I. boliviensis* populations was studied globally for all markers taken together. For the 10 microsatellites, we used exact tests with Markov chains, 10,000 dememorizations parameters, 20 batches, 5000 iterations per batch, and both genotypic and genic frequencies. As a third strategy, we used the Nei’s (1978) genetic distance [83] and the Wright F-statistics [84] with Michalakis and Excoffier (1996)’s procedure [85] for *I. boliviensis* population pairs. The procedure used to measure the significance of F_ST_ were 10,000 randomizations of genotypes among populations and did not assume random mating within populations by means of the log-likelihood G test [86]. Possible theoretical gene-flow estimates among *I. boliviensis* populations studied were measured using the private allele model [87] and from F_ST_ statistics. The fourth strategy was to use the procedure of Ciofi et al. (1999) [88], which was applied to determine if the Bolivian bufeo populations were mainly modeled by gene flow or gene drift by means of the 2mod program. This is based on the genealogical history of alleles between the populations, considering two models of population structure. The gene flow method assumes that the gene frequencies within populations are determined by a balance between gene drift and immigration. The gene drift model assumes that an ancestral panmictic population split into several independent populations which diverged purely by gene drift. It is assumed that the mutation rates are much smaller than the gene flow rates, and that the reciprocal of the mutation rate is much longer than the divergence time in the gene drift model. This method [88] is based on the comparison of the likelihoods for the two models by means of coalescence theory and Markov chain Monte Carlo simulations. Two independent simulations were carried out with 100,000 replicates. For the fifth procedure we developed diverse assignment analyses by using the GENECLASS 2 Program [89] with the microsatellites. We performed different strategies by using Bayesian, frequency, and genetic-distance procedures. The assignation analyses were carried out without simulations and served to estimate the probabilities of individuals belonging to, or being excluded from, the original populations where they were a priori assigned (*p* < 0.05). Some assignation analyses were also completed with 10,000 resampling simulations by means of the Monte Carlo technique and with the procedure of Paetkau et al. (2004) [90]. Additionally, we estimated the possible existence of first-generation migrants in the different *I. boliviensis* populations by using the Bayesian, frequency, and genetic distance procedures we commented on above without simulations. To determine this, we considered the relationship L = L_home_/L_max_. This is the ratio of the likelihood computed from the population where the individual was sampled (L_home_) over the highest likelihood value among all population samples including the population where the individual was sampled (L_max_) [90].

For the *DQB-1* locus, we obtained the average estimate of evolutionary divergence of nucleotide sequence pairs with the Kumar distance (AEDNS) [91], as well as the average estimate of evolutionary divergence among amino acid sequence pairs with the corrected Poisson distribution (AEDAAS) [91]. Additionally, a minimum evolution tree (ME) with the Kumar distance was obtained to determine the relationships among the 11 *DQB-1* alleles found in the Bolivian bufeos and those found in *Inia* populations in the Peruvian Amazon rivers and in the Orinoco (Colombian and Venezuelan) basin rivers.

For all the molecular markers used, we conducted a Bayesian Analysis of Population Structure with the software BAPS v6.0 [92], using mixed and unmixed models to group genetically similar individuals into panmictic genetic clusters. Calculations were performed, with the number of k clusters varying from 2 to 20. Five replicates were carried out for each k value.

#### 2.3.5. Spatial Genetic Structure

Three methods were used for all the molecular markers. First, the Mantel test [93] was used to detect possible relationships between a genetic matrix of the *I. boliviensis* specimens analyzed (Kimura 2P genetic distance) and the geographic distance matrix among the specimens analyzed throughout the Bolivian rivers. In this study, Mantel’s statistic was normalized according to Smouse et al. (1986) [94], which transforms the statistic into a correlation coefficient. Second, a spatial autocorrelation analysis utilized the A_y_ statistic [95] for each distance class (DC). Ay can be interpreted as the average genetic distance between pairs of individuals that fall within a specific DC, with a value of 0 when all individuals within a DC are genetically identical and a value of 1 when all individuals within a DC are completely dissimilar. The probability for each DC was obtained using 10,000 randomizations. For this analysis, which was carried out with AIS v1.0 software [95], five DCs were defined for each one of the molecular markers. The third procedure used was a genetic-landscape interpolation analysis (GLIA), which was performed by means of the “inverse distance weighted” method [95], to view spatial genetic structure of the data in three-dimensional space.

For CR, two additional methods were carried out. First, we created distograms based on Gregorius’s genetic distance and the common shared haplotypes for the specimens of the *Bolivian bufeos* studied [96,97]. Five DCs were used. The significance of distograms and autocorrelation coefficients were calculated by means of 10,000 Monte Carlo simulations [98], with an estimated confidence interval of 95% [99]. We used the SGS v1.0d software to determine the significance of these autocorrelation coefficients [100]. Second, we used AIDA as a second spatial autocorrelation procedure [101] with the *II* and *cc* statistics, using 10 populations for CR. To connect the geographic localities within each distance class, we used the Gabriel–Sokal network [102,103,104] and Delaunay triangulation with elimination of the crossing edges [105]. The Bonferroni and Kooijman tests [106] were applied to determine the significance of the autocorrelation coefficients. Six DCs were used in this analysis.

#### 2.3.6. Possible Historical Demographic Changes

We used three strategies to determine possible female demographic changes in the Bolivian bufeo population for CR. First, to reconstruct the possible relationships among the *I. boliviensis* haplotypes analyzed, to estimate the possible divergence times among these haplotypes, and to determine if the form of a median-joining network (MJN) agreed with population change, we constructed a. MJN using Network v4.6.0s (Fluxus Technology Ltd., Colchester, UK) [107]. Additionally, the ρ statistic [108] and its standard deviation [109] were estimated and transformed into years. The ρ statistic is unbiased and highly independent of past demographic events; furthermore, this “borrowed molecular clocks” approach uses direct nucleotide substitution rates inferred from other taxa [110]. Here, we used an evolutionary rate of 1.5% per one million years [111], representing one mutation each 166,167 years for the CR, which is the same as that used for other cetaceans, including *Inia* [11]. Network analyses can be more useful in the reconstruction of evolutionary history within a species or among closely related species than bifurcating trees. Second, we used Fu and Li’s D* and F* tests [112], Fu’s F_S_ statistic [113], Tajima’s D test [114], and R2 statistic [115]. Both 95% confidence intervals and probabilities were obtained with 10,000 coalescence permutations. Also, a mismatch distribution (pairwise sequence differences) was obtained [116]. We used raggedness (*rg*) to determine the similarity between the observed and theoretical curves. These demographic analyses were carried out using DNAsp v5.1 [59] and Arlequin v3.5.1.2 [60]. Third, we used the coalescence-based Bayesian Skyline Plot (BSP) to estimate demographic changes in female effective numbers. BSP analysis was performed in BEAST v2.4.3 using the empirical base frequencies and a strict molecular clock [117,118]. We applied jModelTest2 to evaluate the best substitution models. Additionally, we assumed a substitution rate of 5 × 10^–8^ substitutions per site and per year to obtain the time estimates in years, as well as with kappa with log-normal (1, 1.25), and skyline population size with uniform (0, infinite; initial value 80). We conducted a total of five independent runs of 40 million Markov chain Monte Carlo (MCMC) iterations following a burn-in of 10% of iterations, logging every 10,000 iterations. We selected a stepwise (constant) Bayesian skyline variant with maximum time as the upper 95% high posterior density (HPD) and trace of the root height as treeModel.rootHeight. The convergence of the analysis was assessed by checking the consistency of the results over five independent runs. For each run, we used the software Tracer v1.7 to inspect the trace plots for each parameter to assess the mixing properties of the MCMCs and to estimate the ESS value for each parameter. Runs were considered as having converged if they displayed good mixing properties and if the ESS values for all parameters were greater than 200. We discarded the first 10% of the MCMC steps as a burn-in and obtained the marginal posterior parameter distributions from the remaining steps using Tracer v1.7. To test whether the inferred changes of effective numbers over time were significantly different from a constant-population-size null hypothesis, we compared the BSP obtained with the ‘Coalescent Constant Population’ model (CONST) implemented in BEAST v2.4.3 using Bayes Factors. Therefore, we conducted five independent CONST runs using 40 million MCMC iterations after a burn-in of 10%, logging every 10,000 iterations. We assessed the proper mixing of the MCMC and ensured ESS values were greater than 200. We then used the Path sampler package in BEAST v2.4.3 to compute the log of marginal likelihood (logML) of each run for both BSP and CONST. We set the number of steps to 100 and used 40 million MCMC iterations after a burn-in of 10%. Bayes Factors were computed as twice the difference between the log of the marginal likelihoods (2[LogML(BSP)—LogML(CONST)]) and were performed for pairwise comparisons between BSP and CONST runs. As recommended, Bayes Factors greater than 6 were considered as strong evidence to reject the null hypothesis of constant effective numbers throughout time. Nevertheless, all these demographic procedures have several caveats. Selection can affect effective population size, reducing the effective number for a time and increasing the coalescence rate later [119]. The same occurs with small changes in the mutation rate (μ), which can greatly affect the effective number and, in turn, estimated divergence time [120]. However, we assume that the mitochondrial control region has complete absence of selective sweeps, and that it has a neutral behavior.

For the nuclear microsatellite markers, five tests were used to determine possible demographic changes. First, the test of Kimmel et al. (1998) [121] is based on an imbalanced β index. If a population is in equilibrium, has a constant demographic size, and is not suffering an expansion, β = 1 (ln β = 0). In contrast, if a population has suffered an expansion coming from a mutation-drift equilibrium situation (constant population size), β < 1 (ln β < 0). If a population has experienced an expansion coming from a previous bottleneck, β > 1 (ln β > 0). This last value will be present for a long time (several thousand generations) before showing the signature of a population expansion, β < 1 (ln β < 0). There is an exception to this general rule, when a bottleneck is so intense that the population becomes monomorphic before the demographic expansion, in which case, β < 1 (ln β < 0) all the time. All these β values are consistent in stepwise, logistic or exponential population growth, and are not especially affected in diverse mutation models [121]. To determine the statistical significance of the β (ln β), a jackknife procedure [122] was applied to obtain the variance of β and, with this variance, a Student’s *t* test at 95 and 99% confidence intervals was used. Second, the test of Zivothovsky et al. (2000) [123] was used to calculate an expansion index: S_k_ = 1 − ((K − (R_k_V/2)/5V^2^). *K* and *V* are the unnormalized kurtosis (fourth central moment) and the allele size variance, estimated from a sample and corrected for sampling bias, respectively. R_k_ = k_m_/σ^2^_m_ (k_m_ = kurtosis and σ^2^_m_ = variance in the repeat-number mutational changes). We used an R_k_ value of 6.3 because it was obtained for dinucleotide microsatellites [124], and because dinucleotide microsatellites were used in the current study. Some authors [125] used the same data and a geometrical distribution of mutational events and obtained an estimated σ^2^_m_ of 2.5. This is basically the same as what was obtained by using a truncated Poisson distribution [123], as well as what we calculated in the present work (σ^2^_m_ = 2.39). The value of S_k_ is expected to be 0 in a general symmetric stepwise mutation model for a population in equilibrium, with constant size [126]. The S_k_ is positive if an expansion affected the population, and it is negative if a bottleneck affected the population. To obtain demographic conclusions of this analysis, the within-population variance and the expansion index are averaged for all the microsatellites studied within each population and their dynamics are compared. It was shown that a significant correlation existed between V and S_k_ (r = 0.58) for a human data set [123]. However, this correlation was moderate and, in fact, both statistics could react differently to changes in population size and have dissimilar patterns across populations. A jackknife procedure was performed to obtain the variance of S_k_ and, with this variance, a Student’s t test was used and a 95% confidence interval was estimated. A third procedure used to detect any possible bottleneck was created by Garza and Williamson (2001) [127]. This procedure is based on the ratio M = k/r, where *k* is the total number of alleles detected in a locus given and *r* is the spatial diversity (that is, the distance between alleles in number of repeats and the overall range in allele size). When a population is reduced in size, this ratio will be smaller than in equilibrium populations. To calculate this M value, the program will simulate an equilibrium distribution of M and give assumed values for three parameters of the two-phase mutation model (θ = 4Neμ, p_s_ = mean percentage of mutations that add or delete only one repeat unit, and Δ_g_ = mean size of larger mutations). Once M is obtained, it is ranked, relative to the equilibrium distribution. Using conventional criteria, there is evidence of a significant reduction in population size if less than 5% of the replicates are below the observed value. The average θ values used in this analysis were obtained from the MISAT program [128]. This analysis was carried out with the M-P-Val and Critical-M programs [127]. The fourth procedure was the locus kurtosis (*k*) test [129], which is based on the following principles. A population with constant size has gene genealogies, which tend to have a single ancient bifurcation. Therefore, the allele length distributions have multiple discrete peaks. In contrast, in a growing population, most of the gene genealogy bifurcations tend to date back to the time expansion and, as a result, the allele length distribution is clearly more smoothly peaked. To assess significance levels, a binomial distribution is used with the number of trials equal to the number of loci based on the expectation of an almost (*p* = 0.515) equal probability of negative and positive k-values for the set of loci analyzed. When there is a smaller loci number associated with positive k values than would be expected for a constant-sized population, there is evidence of a population expansion. The significance level of the binomial distribution was measured using the Statistics Sample Program written by Michael H. Kelly. The fifth procedure is the interlocus (g) test [130]. This test focuses on the following facts. This test shows that for stable populations, the allele size variance is highly variable among loci, whereas in expansion populations this variance is usually lower. Therefore, allele sizes with sufficiently low variances are taken as evidence of population expansion. A low value of g is taken as a sign of population expansion. Significance levels for the interlocus test are found on Table 1 [130], which shows the fifth-percentile cutoffs for the interlocus test. It was noted that this last test is probably the most powerful for this task [131].

## 3. Results

### 3.1. Genetic Diversity in I. boliviensis

The genetic diversity for the CR was composed of 15 haplotypes (with a total of 39 mutations), H_d_ = 0.632 ± 0.056, π = 0.0051 ± 0.0013, and θ_per sequence_ = 7.835 ± 2.296. Two of the haplotypes contained a large fraction of the specimens analyzed, H1 (48/82 = 58.53%) and H2 (13/82 = 15.85%). H1 was spread across all different parts of the Mamoré River sampled, as well as in the Ipurupuru, Ibaré, and Securé rivers. H2 was spread throughout the Ipurupuru, Tijamuchi, and in a determined point of the Mamoré River. Haplotypes H4, H5, H6, H7, H10, and H11 were distributed in determined points of the Mamoré River. H8 and H9 were only found in the Iténez (Guaporé) River. H12 and H13 were only in the San Martín River, and H14 and H15 were only in the Securé River. Therefore, two haplotypes were spread across the entire geographical area analyzed, whilst the others were found at particular locations of the area sampled. These mitochondrial genetic-diversity levels are a medium for the CR found in other mammals [3].

For the nuclear markers, the genetic diversity levels were as follows. For the microsatellites, MNA was 4 ± 1.55 (extreme microsatellites: *EV37* with seven alleles, and *KWM2B* and *EV94* with only two alleles), H_o_ was 0.291 ± 0.256, and H_e_ was 0.345 ± 0.230. These are low genetic-diversity-level values for this kind of markers. The genetic diversity for the *DQB-1* locus showed 11 different alleles (with a total of 19 mutations) with H_d_ = 0.857 ± 0.165, π = 0.031 ± 0.016, and θ_per sequence_ = 3.641 ± 1.231. The three alleles with the highest frequencies were *DQB-1 0101* (14/43 = 32.56%), *DQB-1 0201B* (7/46 = 15.22%), and *DQB-1 0204* (7/46 = 15.22%). The first allele was widely distributed in different parts of the Mamoré River, as well as in the San Martín, and Securé Rivers. The second allele was found in the same rivers as the previous one. plus the Iténez (Guaporé) River. The third allele was present in the Mamoré, Ibaré, and Securé Rivers. The genetic variability at the *DQB-1* locus was medium–high.

### 3.2. Hardy–Weinberg Equilibrium in the I. boliviensis Populations

Two kinds of markers allow us to test the possible existence of HWE in the *I. boliviensis* population: microsatellites and the *DQB-1* locus. For the microsatellites, if we consider the overall sample, there was a significant excess of homozygotes (Fisher’s method: χ^2^ = infinite; 18 df; *p* < 0.00001). Those loci with a significant excess of homozygotes were *EV76* (F_IS_ = 0.346; *p* < 0.00001), *EV96* (F_IS_ = 0.255; *p* < 0.0025), *MK5* (F_IS_ = 0.337; *p* < 0.03), *PPHO137* (F_IS_ = 0.037; *p* < 0.011), and *EV37* (F_IS_ = 0.402; *p* < 0.00001). By population, only two yielded heterozygote deficits: the confluence of the Ibaré and Mamoré Rivers (pop 10) (*p* < 0.005), and Middle Mamoré (pop 1) (*p* < 0.0134). Only two loci contributed to this heterozygote deficit: *EV96* (*p* < 0.016), and *EV37* (*p* < 0.00001). When we considered a hierarchical scheme with the Wright F_IS_ statistic, three out of ten microsatellites presented homozygote excess, *EV76* (*p* = 0.018–0.002), *EV96* (*p* = 0.039–0.021), and *EV37* (*p* < 0.001). However, homozygous excess was not significant when all the loci were analyzed collectively (*p* = 0.187).

There were no deviations from the HWE at the *DQB-1* locus, at the overall population level (*p* = 0.236), or at single-river level (Mamoré River, *p* = 0.488; Iténez-Guaporé River, *p* = 0.201; Securé River, *p* = 1.000).

### 3.3. How Many “Gene Pools” of I. boliviensis Are in the Mamoré–Iténez (Guaporé) River Basin?

#### 3.3.1. Mitochondrial Control Region

The most probable nucleotide substitution models were T92 ± I for BIC (5605.82) and GTR ± I for AIC (4110.39) at the CR. The ML tree for the CR (Figure 2) showed that a major part of the specimens sampled in the core of the area sampled comprised a big cluster, where these specimens were mixed independently of the populations in which they were classified a priori and where the geographical rapids did not play any role. The most differentiated individuals were some of those sampled in the periphery of the sampled area of the Mamoré River and in the Iténez (Guaporé) River (El Corte and El Azul, pop 3 and pop 4, respectively). This analysis showed that the rapids of the Mamoré River had no measurable effect on the geographical structure of *I. boliviensis*. It is interesting to note that *Pontoporia blainvillei* is the sister species of *Inia* and the Asian River dolphin (*Platanista gangetica*) is not closely related to *Inia* for CR. This result is consistent with basically all previous molecular-clock analyses of cetaceans that have been carried out properly [132].

The overall genetic heterogeneity for CR was significant for all the statistical analyses we used (Table 1). However, the amount of gene flow among the considered populations was relatively high, although these statistics yielded significant heterogeneity (N_ST_ = 0.250, *p* = 0.001 and Nm = 1.51; F_ST_ = 0.251, *p* = 0.001 and Nm = 1.49). Based on the exact probability tests, 16 of 45 (35.56%) population pairs showed significant heterogeneity (Table 2). Nevertheless, the population pairs Middle Mamoré–Bolivar (pop 1 and pop 2), Bolivar–El Corte (pop 2 and pop 3), and El Corte–El Azul (pop 3 and pop 4), which are separated by rapids, did not show significant heterogeneities (*p* = 0.713 ± 0.005; *p* = 1.000 ± 0.000; *p* = 0.521 ± 0.006, respectively). The gene flow estimates among population pairs (Table 2) showed that most of the estimates were higher than one, which could be considered elevated values of gene flow (26/45 = 57.78%). Eleven out of nineteen cases pointed to El Azul (pop 4) and San Martín River (pop 5), both being the most distant from the core of the area sampled. The Kimura 2P genetic distances among populations for CR (Table 3) showed small distances. The most-differentiated population was El Azul (pop 4) (2.6–3.5%, with respect to the other populations). All the other population pairs had smaller genetic distances.

A series of AMOVAs were carried out with different population combinations for CR (Table 4). In this case, two populations offered the highest percentage of variation (72.41%) (El Azul-pop 4 vs. all the other populations), followed by three possible populations (65.48%; El Azul-pop 4 vs. San Martín River-pop 5 vs. all the other populations). All the other combinations showed percentages of variation that were considerably lower. Thus, it was clear from the CR data that El Azul was the most differentiated population.

The BAPS analysis is shown in Figure 3. For the CR, the best breakdown was three populations (log (marginal likelihood) = −349.619) using both mixed and unmixed models. One population was integrated with two specimens of the Middle Mamoré (pop 1) (from the locality named Cerrito). Another population was composed of three specimens (two specimens from El Azul, pop 4, plus a specimen from El Corte, pop 3, although these two last populations were separated by the San Jose Rapid). The third population was integrated with all the other specimens. Therefore, the different rapids considered in this study did not contribute to differentiation of *I. boliviensis* populations in Bolivian rivers using CR.

#### 3.3.2. Nuclear Markers

##### Microsatellites

The CFA, considering both individuals and populations, showed that the three main factors explained 72.34% of the variance of the model for the 61 specimens analyzed for microsatellites (Figure 4). Only the specimens of El Azul (pop 4) were separated from all the other specimens. Like that found with the mtDNA, the microsatellites detected specimens from El Azul (pop 4) as the most differentiated.

Using genotypic differentiation with exact G tests, only four out of 28 (14.29%) comparison population pairs for the 10 microsatellites analyzed (Table 5), were significant at the level of α = 0.05. These were the cases of the confluence of the Ibaré and Mamoré Rivers (pop 10 for mtDNA and pop 8 for microsatellites) vs. Middle Mamoré (pop 1) (*p* = 0.0051), confluence of the Ibaré and Mamoré Rivers (pop 8) vs. El Corte (pop 3) (*p* = 0.0008), confluence of the Ibaré and Mamoré Rivers (pop 8) vs. San Martín River (pop 5) (*p* = 0.0395), and Middle Mamoré (pop 1) vs. Securé River (pop 6) (*p* = 0.0355). If we apply the Bonferroni correction (α = 0.0017), only the population-pair confluence of the Ibaré and Mamoré Rivers (pop 8) vs. El Corte (pop 3) is significant. The microsatellites that showed significant genetic heterogeneity were *EV76* (*p* = 0.0004), *EV96* (*p* = 0.0019), *KWM12A* (*p* = 0.0468), and *PPHO137* (*p* = 0.0010). Globally, the set of 10 microsatellites we used showed capacity to significantly differentiate populations (χ^2^ = 58.274, df = 20, *p* = 0.000013).

Using genic differentiation with exact G tests, seven out of 28 (25%) comparison population pairs for the 10 microsatellites analyzed (Table 6) were significant at the level of α = 0.05. This percentage was slightly higher than in the previous case. These were the cases of the confluence of the Ibaré and Mamoré Rivers (pop 8) vs. Middle Mamoré (pop 1) (*p* = 0.0010), the confluence of the Ibaré and Mamoré Rivers (pop 8) vs. El Corte (pop 3) (*p* = 0.00005), the confluence of the Ibaré and Mamoré Rivers (pop 8) vs. El Azul (pop 4) (*p* = 0.00001), El Corte (pop 3) vs. El Azul (pop 4) (*p* = 0.0447), the confluence of the Ibaré and Mamoré Rivers (pop 8) vs. San Martín River (pop 5) (*p* = 0.0048), El Corte (pop 3) vs. San Martín River (pop 5) (*p* = 0.0093), and the confluence of the Ibaré and Mamoré Rivers (pop 8) vs. Securé River (pop 6) (*p* = 0.0189). If we apply the Bonferroni correction (α = 0.0017), only three population pairs are significant. The microsatellites that showed significant genetic heterogeneity were *EV76* (*p* = 0.00001), *EV96* (*p* = 0.0008), *PPHO137* (*p* = 0.0001), and *EV37* (*p =* 0.0035). Globally, the set of 10 microsatellites we used showed capacity to significantly differentiate populations (χ^2^ = infinite, df = 20, *p* = 0.000001). Thus, although some genetic heterogeneity among the bufeo populations was detected with microsatellites, this heterogeneity was relatively low.

The genetic distance [83] among pairs of populations is shown in Table 7. Again, El Azul (pop 4) was the most differentiated population (D_N_ = 0.188–0.072). It had the lowest genetic distance from El Corte (pop 3), the closest geographical population, although the two were separated by the San José rapid. The second more differentiated population was San Martín River (pop 5) (D_N_ = 0.183–0.058), the most isolated population. Rapids did not have a measurable effect on genetic distances among populations.

The F_ST_ and the Nm (gene flow) statistics among population pairs are shown in Table 7 Most comparisons yielded values higher than 1 (elevated gene flow). If we consider a hierarchical scheme with the Wright F_ST_ statistic, only three out of ten microsatellites present significant genetic heterogeneity: *EV76* (*p* < 0.001), *EV96* (*p* < 0.007), and *PPHO13* (*p* < 0.007). However, when random mating is not assumed, and all loci are analyzed together using the log-likelihood G, there was significant heterogeneity (*p* < 0.001).

The 2mod analysis with 100,000 simulations was repeated three times for microsatellites. In each case, 99.9% of the simulations showed that the gene flow model was more important than the gene drift model for the bufeos in the Bolivian rivers analyzed. Thus, the Bayesian factor in favor of the gene flow model was 999. This means that the gene flow inside of the Bolivian rivers is an evolutionary factor of extreme importance among the *I. boliviensis* populations. The gene flow among bufeo populations indicates that rapids are of little importance in acting as genetic barriers for these populations. In fact, the gene flow using the private-allele method was 2.034, which could be considered a high value.

The analyses with GeneClass 2 generally showed low levels of correct assignation, with the lowest percentage of 37.7% (23/61, with the criterion of Nei’s 1972 genetic distance [133] and the simulation algorithm of Paetkau et al., 2004 [90]) and the highest percentage of 50.8% (31/61, with the criterion of Paetkau et al., 1995 [134] and the simulation algorithm of Rannala and Mountain, 1997 [135], as well as the criterion of Cavalli-Sforza and Edwards, 1967 [136] and the simulation algorithms of Cornuet et al., 1999 [137] and Rannala and Mountain, 1997 [135]). These results showed that the similarity among the considered *I. bolivienis* populations is high. The results also indicate that there was insufficient microsatellite differentiation to correctly link most of the specimens studied to the populations or localities where they were sampled. Indeed, the same program was used to detect the first generation of migrants in the Bolivian rivers analyzed. The criteria of Nei [133] and Cavalli-Sforza and Edwards [136] and the genetic distances calculated with the simulation algorithms of Cornuet et al. (1999) [137] and Rannala and Mountain [135] did not detect first-generation migrants. Nevertheless, other criteria with other simulation algorithms detected 12 first-generation migrants (*p* = 0.01; the criterion of Baudouin and Lebrun, 2000 [138] with the simulation algorithm of Rannala and Mountain [135], as well as the criterion of Paetkau et al., 1995 [134] with the simulation algorithm of Cornuet et al., 1999 [137]), or up to 14 first-generation migrants (*p* = 0.01; the criterion of Baudouin and Lebrun, 2000 [138] with the simulation algorithm of Cornuet et al., 1999 [137]). As an example, four first-generation migrant individuals (*p* = 0.01) were detected with the criterion of Rannala and Mountain [135] and the simulation algorithm of Paetkau et al. (2004) [90]. They were specimen Bv14-sampled in the confluence of the Ibaré and Mamoré Rivers (pop 10 for mtDNA and pop 8 for microsatellites) but coming from the upper Ibaré River (pop 7) (*p* = 0.005), specimen B15 sampled in Bolivar (pop 2) but coming from El Corte (pop 3) (*p* = 0.006), specimen F01 sampled in the San Martín River (pop 5) but coming from Middle Mamoré (pop 1) (*p* = 0.00001), and specimen F03 sampled in the Securé River (pop 6) but coming from Middle Mamoré (pop 1) (*p* = 0.01). These data show that, for nuclear microsatellites, neither geographical barrier (rapids) nor distance is an obstacle to the dispersion of the bufeos within Bolivian rivers.

The BAPS analyses for microsatellites are shown in Figure 5. When we only considered the 61 individuals analyzed without possibility of admixture, there were six different populations (log (marginal likelihood) = −707.4; *p* = 0.859). The first cluster contained 25 specimens from the confluence of the Ibaré and Mamoré Rivers (pop 8) (17 specimens), Middle Mamoré (pop 1) (3 specimens), Bolivar (pop 2) (2 specimens), El Corte (pop 3) (1 specimen), and the Securé River (pop 6) (2 specimens). The second cluster had seven specimens from the confluence of the Ibaré and Mamoré Rivers (pop 8) (five specimens), and Middle Mamoré (pop 1) (two specimens). The third cluster contained 15 specimens from the confluence of the Ibaré and Mamoré Rivers (pop 8) (one specimen), Ibaré River (pop 7) (three specimens), Middle Mamoré (pop 1) (five specimens), El Corte (pop 3) (two specimens), El Azul (pop 4) (1onespecimen), San Martín River (pop 5) (two specimens), and the Securé River (pop 6) (one specimen). The fourth cluster consisted of three specimens from the confluence of the Ibaré and Mamoré Rivers (pop 8). The fifth cluster only had one specimen from the confluence of the Ibaré and Mamoré Rivers (pop 8). Cluster six had ten specimens from the confluence of the Ibaré and Mamoré Rivers (pop 8) (three specimens), Middle Mamoré (pop 1) (two specimens), Bolivar (pop 2) (one specimen), El Corte (pop 3) (three specimens), and El Azul (pop 4) (one specimen).

When the admixture model among the individuals was used, all the specimens showed different degrees of mixing. When the analysis was carried out considering the eight populations and no admixture, then BAPS detected three clusters as the most optimal partition (log (marginal likelihood) = −754.6; *p* = 1). The first cluster contained all the individuals from the confluence of the Ibaré and Mamoré Rivers (pop 8). The second cluster had all the individuals from Middle Mamoré (pop 1), Bolivar (pop 2), and El Corte (pop 3). Cluster three had all the individuals from Ibaré River (pop 7), El Azul (pop 4), San Martín River (pop 5), and the Securé River (pop 6). When the analysis was carried out with populations and with the admixture model, only one cluster was detected. This means that, for microsatellites, the entire area analyzed should be consider as a unique population.

##### DQB-1 Locus

The AEDNS was 0.017 and the AEDAAS was 0.064. Figure 6 displays the relationship among the 11 *DQB-1* alleles found in the Bolivian rivers and those found in *Inia* populations from the Peruvian Amazon rivers and those of the Orinoco (Colombia and Venezuela) basin. The ME tree with the Kumar distance showed that only five alleles were exclusive to the Bolivian bufeos, but they did not show any geographical structure. Another noteworthy result was the fact that four *DQB-1* alleles were shared among bufeos from Bolivia, Peruvian Amazon, and the Orinoco basin, whilst two *DQB-1* alleles were shared between Bolivian and Peruvian Amazon rivers. This strongly disagrees with the fact that the Bolivian mitochondrial haplotypes were private and exclusive, and not shared with the bufeos of other river basins.

The BAPS analyses of the *DQB-1* locus (23 specimens, 46 alleles; Figure 7) detected four different clusters (log (marginal likelihood) = −250.9; *p* = 0.998). The first cluster contained twenty-six alleles found in the Middle Mamoré (pop 1) (seven alleles), Bolivar (pop 2) (one allele), El Corte (pop 3) (five alleles), El Azul (pop 4) (two alleles), the confluence of the Ibaré and Mamoré Rivers (pop 8) (six alleles), San Martín River (pop 5) (three alleles), and the Securé River (pop 6) (two alleles). The second cluster contained ten alleles from Middle Mamoré (pop 1) (three alleles), confluence of the Ibaré and Mamoré Rivers (pop 8) (four alleles), Ibaré River (pop 7) (two alleles), and the Securé River (pop 6) (one allele). The third cluster was composed of seven alleles from Middle Mamoré (pop 1) (two alleles), Bolivar (pop 2) (one allele), El Corte (pop 3) (one allele), confluence of the Ibaré and Mamoré Rivers (pop 8) (one allele), San Martín River (pop 5) (one allele), and the Securé River (pop 6) (one allele), while the fourth cluster consisted of three alleles from Middle Mamoré (pop 1) (two alleles), and the confluence of the Ibaré and Mamoré Rivers (pop 8) (one allele). As with what we observed with the previous nuclear markers, the rapids we analyzed did not cause any detectable differentiation among the *I. boliviensis* populations considered.

### 3.4. Spatial Genetic Patterns in Inia boliviensis

#### 3.4.1. Mitochondrial DNA

The log Mantel’s tests applied to the CR (Figure 8) were significant, showing that the geographical distances explained 13.21% of the genetic distances (r = 0.365; *p* < 0.00009). For the mitogenomes, the geographical distances explained 10.73% of the genetic distances (r = 0.328; *p* < 0.00009). Thus, this test globally detected evidence of isolation by distance for the mtDNA. Also, using individuals, the spatial autocorrelation analysis with the A_y_ statistic showed for the CR (Figure 9), using five DCs, an overall correlogram significant (V = 0.00321, *p* < 0.0008). For the first and second DCs, significant positive autocorrelation was found (1 DC: 0–12.43 km, *p* < 0.000001; 2 DC: 12.43–22.38 km, *p* < 0.0003), whilst the fourth and fifth DCs yielded significant negative autocorrelations (4 DC: 58.33–106.19 km, *p* < 0.0089; 5 DC: 106.19–284.27 km, *p* < 0.0024). Thus, this correlogram with individuals shows monotonic clines which agree with isolation by distance. The spatial autocorrelation studied with populations (10) by means of AIDA, for CR, with the Moran’s *II* index and with six DCs indicated significance for distance classes 1, 2, 4, 5, and 6. The first DC was positive (0–28.28 km; *II* = 0.0754, *p* < 0.005). The second, fourth, fifth, and sixth DCs were all negative (2 DC: 28.28–50.18 km, *II* = −0.0761, *p* < 0.005; 4 DC: 111.80–145.17 km, *II* = −0.0502, *p* < 0.01; 5 CD: 145.17–261.99 km, *II* = −0.0480, *p* < 0.05; 6 DC: 261.99–495.60 km, *II* = −0.0666, *p* < 0.005). Thus, the Moran’s *I* index detected a significant genetic patch of around 30 km in diameter. Later, an isolation-by-distance model was detected. With the AIDA procedure with the Geary’s *cc* coefficient and six DCs, two DCs were significant. The first DC was significantly positive (0–28.28 km; *cc* = 0.3445, *p* < 0.005) and the sixth DC was significantly negative (261.99–495.60 km; *cc* = 2.0804, *p* < 0.05). As in the previous case, an isolation-by-distance model explains the spatial pattern found for *Inia* in the Bolivian rivers analyzed. However, when the spatial structure was studied using haplotypes (Figure 10) and distograms for CR with five DCs both with the Gregorious (1978) distance and with haplotypes in common, no significant spatial structure was detected. It is interesting to note that when individuals and populations were used to determine spatial structure, it was clearly found for mtDNA. In contrast, when haplotypes (and not individuals nor populations) were used, no spatial structure was detected.

The GLIA for CR showed that the population of San Martín River (pop 5) was the most differentiated of the populations studied (Figure 11).

#### 3.4.2. Nuclear Markers

##### Microsatellites

The spatial patterns found with microsatellites totally disagree with that found with mtDNA. For example, the Mantel test detected a negative relationship between the geographic and the genetic distances (r = −0.083; *p* < 0.0227) (Figure 12). The overall correlogram with the A_y_ statistic and five DCs, using individuals, was significantly significant (V = 0.0344, *p* < 0.0001). However, this correlogram was the opposite to that found with the mtDNA (Figure 13). The first two DCs were significantly negative; the closest geographical individuals were the most genetically differentiated (1 DC: 0–13.31 km, *p* < 0.0001; 2 DC: 13.31–36.27 km, *p* < 0.0002). The third and fourth DCs were significantly positive (3 DC: 36.27–66.12 km, *p* < 0.003; 4 DC: 66.12–113.78 km, *p* < 0.002). The last CD was not significant. The GLIA was very different to that found with the mtDNA. The most differentiated individuals were in the confluence of the Ibaré and Mamoré Rivers (pop 8) (Figure 14).

##### DQB-1 Locus

The spatial pattern at this nuclear locus was different to that determined with all of the other genes we analyzed. The Mantel test did not reveal any relationship between the geographical and genetic distances (r = 0.034; *p* = 0.183). The overall correlogram was not significant (V = 0.00260, *p* = 0.186). The correlogram yielded a crazy-quilt pattern, with no significant DC. Related to this, the GLIA showed multiple peaks and valleys, without a clear spatial pattern.

### 3.5. Evolutionary Demographic Changes in Inia boliviensis

#### 3.5.1. Mitochondrial DNA

With the MJN procedure, and using the CR data (Figure 15), the split between *Inia* and *Pontoporia* was dated between 15.121 ± 0.154 and 13.170 ± 0.138 MYA. This temporal divergence between both genera (15–13 MYA) is very like that obtained in other studies. From a paleontological point of view, at least, a minimal temporal split between the ancestors of *Inia* and *Pontoporia* was estimated to be around 13–11 MYA [139]. From a molecular perspective, three different studies with different procedures yielded a time between 17.6–8.8 MYA [3], 13 MYA [140], and 17–15 MYA [132]. Thus, the time split obtained between both genera was very similar among all these different studies. However, the proliferation of mt haplotypes in the Bolivian rivers analyzed occurred much more recently. The time-split between the two haplotypes which contained most individuals was estimated to have occurred 130,700 ± 42,130 YA (H1, with 48 individuals, and H2, with 13 individuals). The time-split among H1 and all the other haplotypes detected in the Bolivian rivers was estimated to have occurred 180,300 ± 38,800 YA. Therefore, the current haplotype radiation in *I. boliviensis* was estimated to have occurred in the last phase of the Pleistocene. The star form of the MJN supports population expansion.

For the CR data set, all the demographic statistics used showed a significant population expansion for *I. boliviensis* (Tajima’s D = −2.363, *p* < 0.000001; Fu and Li’s D* = −2.613, *p* < 0.012; Fu and Li’s F* = −2.609; *p* < 0.015; Fu’s F_S_ = −5.113, *p* < 0.025; and R2 = 0.027, *p* < 0.000001). In contrast, the mismatch distribution was not significant for CR, and thus no demographic changes were detected with this procedure. The BSP analysis with the CR data set was carried out taking into consideration two different times (Figure 16). Taking the last 0.4 MYA, no clear changes were determined until around 60,000 YA, when a female population decrease began. Around 20,000 YA, this decrease was very strong. Another BSP analysis indicated that this decrease occurred up until around 500 YA, when there was a population increase.

#### 3.5.2. Microsatellite Markers

The imbalance index offered a value of β (= 2.117; t = 5.089, *p* < 0.05) (and ln β = 0.749) significantly higher than 1 (β) or 0 (ln β), thus indicating later population growth from an initial bottleneck. The test of Zhivotovsky et al. (2000) [123] showed a different picture, indicating that each one of the tests used had a different power to detect diverse demographical changes at different historical moments. This test detected a strong and significant bottleneck event in the Bolivian population (S_k_ = −2.253; t = −5.274, 11 df, *p* < 0.01; 99% confidence interval (−3.351, −1.155)). This significant negative S_k_ was accompanied by a very low within-population variance (V = 0.9952 ± 0.2647).

The test of Garza and Williamson [127] agrees quite well with the results shown by the previous test. The Bolivian population showed evidence of a bottleneck event, although not drastic. To carry out this test, we first estimated the θ (= 4N_e_μ) statistic and the percentage of multiple-step mutations, which were θ = 2.227 ± 1.846 and 13.18% for the Bolivian dolphin sample. The average observed M value was 0.7846. A 10,000-simulation run with these parameters (values cited above) and the sample size used resulted in an average M of 0.8143 with 95% of the individual M values for each marker under the critical value of M_c_ = 0.7094 (indicative of a demographic-constant population). Five markers showed an individual M below this critical value (*EV14*, M = 0.6667; *EV76*, M = 0.60; *EV3*7, M = 0.7072; *EV94*, M = 0.3333; and *MK5*, M = 0.6667), and, therefore, this fraction was statistically significant (χ^2^ = 4.77, 1 df, *p* < 0.05).

The k and g tests differed from the previous analyses. Out of 10 loci analyzed, six (*EV76*, *EV14*, *EV37*, *EV94*, *MK5*, and *PPHO137*) showed negative values of k. Both, k (*p* = 0.341) and g (*p* = 0.678) yielded no evidence of significant population expansions for the Bolivian river dolphins.

## 4. Discussion

Few studies have focused on the impact of the rapids and waterfalls in bufeos in general and in *Inia boliviensis* in particular. The first studies of bufeos in Bolivia [1] consisted of informal surveys of various waterways, descriptions of behavior, and preliminary population-density estimates of bufeos in the Ibaré River. The authors concluded that the precipitous decline in population size was because of anthropogenic influences. They also concluded that the Bolivian bufeo was a separate species (*I. boliviensis*) from the animals of the rest of the Amazon and Orinoco basins (*I. geoffrensis*). They determined significant differences at the morphological and morphometric levels and identified two significant morphometric differences that are fundamental to understanding the possible origins of the bufeos. *I. boliviensis* has 33 teeth on each side of its jaws, whereas *I. geoffrensis* has 26–27 teeth and the average neurocranial volume is significantly less in *I. boliviensis* (558 cm^3^) than in *I. geoffrensis* (665 cm^3^). Following Williston’s law, the Bolivian bufeo should have a more primitive form than the current Amazon bufeo. These authors also claimed that an allopatric speciation process (vicariance barrier) split the *Inia* forms. There is a 350 km section of the Madeira River, from Guayaramerín (Mamoré River, Bolivia and Brazil) to Porto Velho (Madeira River, Brazil), which contains 18 rapids and a collective drop of 60 m due to waterfalls such as Teotônio and Jirau [141]. Two of these rapids were formally designated as waterfalls, due to accentuated differences in mean water-surface elevation above and below the rapids [141]. The 900 m-wide Teotônio waterfall, which is the largest of the falls, fell 30 m over a span of 600 m. The Jirau waterfall, the second largest, is upstream of the Teotônio waterfall. It is 730 m wide, spans 1100 m [141], and allegedly split the *Inia* species. Most authors follow [1] in supposing that the Teotônio waterfall delimits the geographic distribution of both species. Nonetheless, it was suggested that during the rainy season bufeos can cross the upper Madeira River rapids (including the Teotônio waterfall) [7]. More recently, two studies [10,11] analyzed the question whether these rapids and waterfalls in Brazil actually separate the forms of *Inia*. The first study analyzed mtDNA of 125 specimens of bufeos from the Madeira River basin sampled upstream and downstream of the 18 rapids on the upper Madeira River (in Brazil) [10]. The authors observed that all individuals of this river up to almost the mouth of the Madeira River belonged to the species *I. boliviensis,* save those near the population of Borba—870 km downstream of the Santo Antonio rapids and 890 km downstream of the Teotonio waterfall. The authors observed that gene flow was always upstream-to-downstream and the mtDNA of *I. boliviensis* invaded at least half of the Madeira River. To study the impact of the Brazilian rapids on bufeos, the second study analyzed nuclear microsatellite markers of bufeos both upstream and downstream of the Brazilian rapids [11]. They determined that there were likely two biological groups throughout the Madeira River, corresponding to the two species of bufeo, *I. boliviensis* (from the Mamoré River and Madeira River to the locality of Borba) and *I. geoffrensis* (from Borba to the mouth of the Madeira River with the Amazon River). Nevertheless, within *I. boliviensis*, they detected two other groups (one between the Mamoré River and upstream of the Teotonio waterfall, and other between this last waterfall and downstream of the Madeira River). These authors determined that the pattern and direction of gene flow observed for microsatellites reflected that of the mtDNA. Between the Mamoré River and the Brazilian rapids region, gene flow was 9.83 in the downstream direction and 0.00 in the upstream direction. Between the rapids and the downstream localities, gene flow was 0.67 in the downstream direction and 0.19 in the upstream direction. Therefore, these rapids did not represent a complete physical barrier to movement and gene flow of bufeos.

The main finding of this work was that the internal rapids of the Mamoré River (Bolivia) did not differentiate the bufeo populations, in accordance with the two studies in Brazil [10,11]; however, isolation by distance could play some role in the differentiation of the populations, at least, for maternal markers such as mtDNA.

### 4.1. Genetic Diversity and HWE in the Bolivian bufeos

The multiple genetic-diversity statistics we calculated for the diverse molecular markers we used offered a range of diversities. For the CR, the diversity was in the medium range, with some haplotypes dispersed across all the rivers studied (H1, H2) whereas other haplotypes were only specific to particular rivers. Ruiz-García et al. (2018) [9] estimated the genetic diversity of the bufeos in different basins of South America, including Bolivia. That study had a smaller sample size for the dolphins from the Bolivian rivers than in the current study. Some of the genetic diversity estimates of that study [9] were lower than those shown in the present study (six haplotypes vs. fifteen haplotypes; H_d_
*=* 0.52 vs. 0.63; π = 0.0014 vs. 0.0051; θ_per sequence_ = 1.08 vs. 7.83, respectively). However, in both cases, the Bolivian estimates were considerably lower, especially for the nucleotide diversity, than those obtained for the bufeos of other river basins (Orinoco basin; 9 haplotypes; H_d_
*=* 0.79; π = 0.025; θ_per sequence_ = 7.7; Amazon basin: 23 haplotypes; H_d_
*=* 0.71; π = 0.012; θ_per sequence_ = 12.42). This is the first evidence that the Bolivian dolphins were affected by some evolutionary event that diminished its genetic diversity in reference to other *Inia* populations.

Nuclear microsatellites also yielded lower levels of genetic diversity for the *Bolivian bufeos* (MANA = 4; H_e_ = 0.345) than for bufeos from the Peruvian Amazon (MANA = 6.2; H_e_ = 0.591) or the Orinoco basin (MANA = 4.8; H_e_ = 0.524) [4]. Other Amazon bufeo populations had higher genetic diversities than did the Bolivian population. In Brazil, H_e_ = 0.59 was reported for two bufeo populations in the Negro River (Ariaú and Novo Airão, Brazil) [142], and, also in Brazil genetic diversities of H_e_ = 0.54 and 0.57 for the Mamirauá Reserve and Tefé Lake, were found, respectively [143]. Thus, for mtDNA and nuDNA (microsatellites), the *Bolivian bufeos* showed significantly less genetic diversity than the *Inia* populations from other South American river basins. We must recall that genetic diversity has been treated by the IUCN as a priority parameter for measuring the risk of extinction of a species, because it represents the substratum on which natural selection acts to promote the adaptation of a species to changing environments. The loss of genetic diversity can lead to loss of reproductive fitness and to a decrease in the evolutionary potential of a given species [144]. Therefore, this parameter is crucial in helping us to determine if conservation measures are warranted for the *Bolivian bufeos*. In contrast, the diversity for the *DQB-1* locus was medium–high, with the three most-frequent alleles well dispersed in all the Bolivian rivers studied (11 alleles; H_d_ = 0.857; π = 0.031). The diversity for the *Bolivian bufeos* for this marker, in contrast to the other markers studied, was like other *Inia* populations in the Peruvian Amazon (11 alleles; H_d_ = 0.873; π = 0.038) and Orinoco (9 alleles; H_d_ = 0.908; π = 0.054) (Ruiz-García, unpublished). Therefore, the *DQB-1* locus has a different evolutionary dynamic in *Inia* than the other three kinds of markers we used.

For microsatellites, a significant excess of homozygotes was detected when all the Bolivian rivers we studied were analyzed collectively. This result could be due to genetic differentiation among certain populations of some rivers (the Wahlund effect). Nevertheless, when each geographical population was individually analyzed, only two out of eight deviated from HWE. This finding could be due to endogamy within these two populations or that there was a Wahlund effect within them. But, in most of the geographical populations considered, there were no deviations from the HWE. It is very likely that, for the *DQB-1* locus, deviations from the HWE were not revealed. This agrees quite well with the fact that all the rivers we analyzed contained a unique major genetic pool for the Bolivian bufeos.

### 4.2. Genetic Heterogeneity

The phylogenetic trees and the genetic heterogeneity statistics for CR indicated that the rapids in the Mamoré River are not geographical obstacles for the dispersion of the Bolivian bufeos, even though these rapids are characterized by relatively fast water, deep channels, and steep banks, with little suitable habitat for *Inia*. However, there was evidence of isolation by distance where the rapids did not play any significant role. With CR, the genetic heterogeneity was created by the two most distant populations (El Azul in the Iténez-Guaporé River and the San Martín River) from the nucleus of the study area. Although the genetic heterogeneity detected was significant, the values of gene-flow estimates were high. The Kimura 2P genetic distances were small or very small among geographical population pairs, with the highest genetic distance for El Azul (the Iténez-Guaporé River) for CR. The same occurred for AMOVAs. The highest differentiation was between El Azul and the remaining populations. Using BAPS, three groups were detected with CR: two small populations (constituting a very-restrictive number of individuals and thus highly affected by gene drift). All the remaining populations were considered as one.

The nuclear microsatellites showed similarities and differences with that found with mtDNA. The most relevant similarities were as follows: (1) the existence of significant limited heterogeneity, mainly caused by the most differentiated population El Azul (Iténez-Guaporé River) being the second most differentiated population with respect to that from San Martín River (Iténez-Guaporé river basin). This is identical to what was determined with CR. (2) Gene flow among geographical populations is also high (2-mod analysis detected that gene flow is 99.9% more probable than gene drift), producing low levels of correct population assignment (up to 12–14 individuals were identified as migrants of first generation). (3) If we consider non-admixed models, the BAPS analyses showed six different groups for individuals and three different groups for populations. But even in this last case, the three populations were separated by the Matacaré, Warnes, Mayo-Mayo, and Envidia rapids, yet they were considered as a single unique population because the geographical barriers did not seem to affect the genetic structure of the bufeos of this area. Thus, as with the mtDNA, the rapids did not play any role in the heterogeneity of these river dolphins in the Mamoré River. If we considered mixed models, only one unique gene pool was determined, indicating the non-significance of the rapids in shaping the genetic structure of this species in the Bolivian rivers. The main difference regarding the mtDNA is that the nuclear microsatellites did not detect the same strong and significant isolation by distance, as detected by the mtDNA. This difference can be explained by the differential gene flow of females and males. It is interesting to discuss the different results obtained with mitochondrial and microsatellite markers. The mt markers seem to be especially influenced by their neutrality and of the dynamic movement of the females and the microsatellites seem to be of neutral nature and clearly influenced by male movements. In reference to the *DQB-1* locus, six alleles were shared with *Inia* specimens from the Colombian–Venezuelan Orinoco and the Peruvian Amazon, while five alleles were exclusive to the Bolivian rivers. However, these five alleles did not show any kind of genetic structure. The fact that some *DQB-1* alleles were shared with other river basins from other countries, while the mtDNA haplotypes were unique (private) to the Bolivian rivers, is indicative of the fact that the first alleles were probably submitted to similar natural-selection pressures by microorganisms, pathogens and parasites in different rivers of South America. The mtDNA haplotypes showed a much more neutral dynamic. BAPS analysis determined four groups at the *DQB-1* locus, but in no case did the rapids show any influence in the groups detected. Henceforth, for the mtDNA data set and for the two kinds of nuclear markers we used, the rapids of the Mamoré River did not play an important role in the genetic structure found in the bufeos of the Bolivian rivers.

Our results can be compared with those obtained by other authors [10]. They estimated relatively low F_ST_ values for mtDNA, but they indicated that the haplotypes had non-random distribution (F_ST_ = 0.169; *p* < 0.001) between the upstream and downstream groups of *I. boliviensis* separated by these Brazilian rapids. In our case, we detected values of F_ST_ slightly higher (F_ST_ = 0.25 for CR), but the estimates of gene flow were elevated sufficiently to connect the populations of dolphins on both sides of the rapids. Henceforth, they concluded that the rapids do not delimit the distribution of *I. boliviensis* upstream and *I. geoffrensis* downstream of the rapids, as previously hypothesized. Our results agree well with the fact that rapids do not influence the genetic structure of the species studied. It should be noted that some of the rapids in Brazil are considerably larger than those tested in Bolivia. However, these authors [10] determined the existence of gene flow using IMa2 and concluded that gene flow was uni-directional, moving from an upstream to downstream direction. In contrast, we determined the existence of gene flow in both directions for all the rapids analyzed in the Mamoré River in Bolivia. This finding is probably because these rapids were smaller than those in Brazil. In this last country, it was found that 15 haplotypes were restricted to the upstream group while 24 haplotypes were restricted to the downstream group, with only two haplotypes shared between the two groups [10]. Excluding rare haplotypes, singletons recently appeared via a mutational process and did not yet have time to expand geographically from their area of origin. We observed the same in Bolivia, with some haplotypes extensively distributed in the different Bolivian rivers we analyzed. Other haplotypes—those that were generated more recently—were restricted to specific points of the area studied. The Brazilian researchers [10] observed bufeos moving across the greatest of the Brazilian rapids in the Madeira River during high-water season, a time when several fish species have their annual spawning migrations [145]. Most of the rapids are not submerged during the dry season and some are not even submerged during the rainy season. Additionally, these Brazilian authors [10] commented that during periods of extremely high water, such as what occurred in 1996, 2006, 2008, 2009 and 2012, all rapids become submerged, permitting unhindered movement of animals. Although extremely high water potentially facilitates the bi-directional movement across the rapids for the bufeos, it also increases the velocity of the river. The deep channels with steep banks and high-velocity current characteristic of the rapids’ region become even more of a barrier for upstream movement of dolphins. Yet, the same phenomenon probably facilitates downstream movement and gene flow of this species. Periods of extremely high waters, therefore, most likely facilitate uni-directional connectivity between the upstream and downstream groups of river dolphins. In the Bolivian rivers, we observed bi-directional movement across the rapids even in the dry period, because the rapids are not as big as some from the Madeira River. For this reason, the gene flow in the Bolivian rivers is bi-directional across the rapids. These bufeos are capable of sustained swimming in fast waters for short periods of time, after which they need to recuperate [146]. The animals use short bursts of energy to cross the rapid, followed by a period of rest in an area with little or no current [10].

Another work [11] also examined microsatellite markers, when these Brazilian authors studied the impact of the Brazilian rapids on bufeos upstream and downstream. Using STRUCTURE, they determined that the most likely number of biological groups throughout the Madeira River was two, corresponding to the two species of bufeo, *I. boliviensis* (from the Mamoré River and Madeira River to the locality of Borba) and *I. geoffrensis* (from Borba to the mouth of the Madeira River with the Amazon River). A principal component analysis separated out specimens from upstream of the rapids (above Guajará-Mirim, Brazil, or Guayaramerin, Bolivia), between the rapids’ region upstream of the Teotônio waterfall (between Fortaleza do Abunã and the Teotônio waterfall), and between the Teotônio waterfall and the Santo Antônio rapids. All these specimens had mtDNA typical of *I. boliviensis*; however, the Teotônio waterfall separates the nuclear genomes of *I. boliviensis,* upstream, and *I. geoffrensis,* downstream. Thus, this barrier is partially porous, allowing low levels of unidirectional connectivity, upstream to downstream. A principal component analysis also separated out specimens (considered *I. boliviensis* × *I. geoffrensis* hybrids in the Madeira River) downstream of all the rapids and specimens from the Mamirauá Reserve on the Japurá River (used as control specimens of *I. geoffrensis*). Two specimens sampled below the Teotônio waterfall were genetically assigned to the between-rapids group. Most likely, these were first-generation migrants, which means that they could cross this geographical obstacle. Nevertheless, using the BayeAss 3 software, it was found they had the highest migration rates within each group, but also relatively high migration rates between the region upstream of all the rapids, and the rapid region upstream of the Teotônio waterfall, and between the downstream region and rapid region downstream of the Teotônio waterfall. Our results showed that the different rapids present within the Bolivian Mamoré River have less impact on the genetic structure of *I. boliviensis* than rapids and waterfalls in the Brazilian territory, because microsatellites showed that the Bolivian rapids neither decreased nor restricted the gene flow to the contiguous populations. For instance, El Corte and El Azul (the most differentiated of all populations studied with microsatellites) are separated by the San José rapid. However, El Corte was more closely related to El Azul than any other population, although they are separated by an important rapid area. Furthermore, microsatellites showed that gene flow was bi-directional (as occurred with mtDNA) between populations separated by rapids in Bolivia. Therefore, the power of the Mamoré River does not affect the ability of the bufeos to cross the rapids in either direction. This finding is different from what was detected for the Madeira River. These differences are reflected in the BAPS analysis, assuming admixture (the real situation), which only detected one unique population of bufeos in the Bolivian rivers and multiple different groups in the Madeira River (STRUCTURE, principal component analysis). This Bolivian “gene pool” is the same as that the Brazilian authors [11] detected upstream from the Brazilian rapids. Another interesting point is that the bufeos belonging to different river basins in Bolivia could be closely related. This is the case of the close genetic relationship between the bufeos of the confluence of the Ibaré and Mamoré Rivers area and those at the Ipurupuru River. The Ipurupuru River, although it belongs to the Iténez-Guaporé river basin, is geographically near the confluence of the Ibaré and Mamoré Rivers. Therefore, in high-water times, when the forest floods, bufeos can move from rivers of different basins, allowing genetic homogenization of bufeo populations across these rivers. Another difference between the Brazilian works [10,11] of Gravena et al. and our study is that they observed a much greater genetic differentiation among the groups determined above, between, and down-stream of rapids for microsatellites, than for mtDNA. In contrast, we found a greater and clearer genetic heterogeneity in the Bolivian rivers for mtDNA (CR: G_ST_ = 0.264; F_ST_ = 0.251) than for microsatellites (G_ST_ = 0.122; F_ST_ = 0.094; R_ST_ = 0.063). Our results are consistent with females staying near reproductive areas (female philopatry), whereas males disperse more widely, in search of breeding opportunities in faraway areas. It may also be possible that females inhabiting areas with powerful rapids [10,11] are moving more actively than males looking for good places to breed or feed. An alternative hypothesis is that the mtDNA showed more ancestral patterns of colonization when rapids and waterfalls of the current Bolivian rivers were higher or more prominent during Pleistocene dry periods. Furthermore, microsatellites showed patterns of more-recent colonization processes. Clearly, the current rapids from the Bolivian rivers are minor and less powerful (less of a geographical barrier to gene flow) than the Brazilian rapids and waterfalls (a bigger barrier to gene flow). In fact, our study’s higher genetic heterogeneity for mtDNA than for microsatellites agrees quite well with that reported in other studies with cetaceans and disagrees with the results of the Brazilian authors [10,11]. In the North American beluga whales, for example, the γ_ST_ values (0.047–0.072) for microsatellites were significantly lower than the γ_ST_ value obtained for the CR (0.409) [147]. Similarly, it was found that female porpoises were more philopatric than males [148], and it was also shown that there was a highly male-biased dispersal among *Phocaena dalli* populations [149], as we found for *I. boliviensis*.

The second Brazilian work [11] considered that *I. boliviensis* can be divided into two distinct management units (MUs) [150] based on mt and microsatellite markers. In the Bolivian area we studied, however, only one MU should be proposed as being one of the MUs detected by these authors [11]. However, other potential MUs could exist in other areas of the Bolivian rivers which were not included in our study. Although some studies have strongly emphasized that *I. boliviensis* was a separate species from *I. geoffrensis* [3,4,8], the results of [10,11], as well as those from [9], raise doubts as to this idea, at least based on the Biological Species Concept (BSC) [151,152]. The mtDNA genome of *I. boliviensis* is in *I. geoffrensis* occurring in the Madeira River below the Teotônio waterfalls, and the nuclear DNA genome of *I. geoffrensis* have introgressed into *I. boliviensis* in the Teotônio waterfalls. This shows that pre-zygote reproductive-isolation mechanisms were not developed between *I. geoffrensis* and *I. boliviensis* as we would expect if the BSC were applied. Some authors [11] affirmed this by stating that “While the two species do exist (*I. geoffrensis* and *I. boliviensis*), there is an extensive hybrid zone in the Madeira River. The hybrid zone appears to be ancient and is characteristic of a region of introgressive hybridization. There appears to be no physical barrier between the hybrids and *I. geoffrensis*, bringing into question how and why the hybrids persist”. The question is easily answered using the BSC: they are a unique bufeo species.

### 4.3. Spatial Genetic Structure

The different types of molecular markers we used in *I. boliviensis* allowed us to detect different spatial genetic structures. For the mtDNA, Mantel’s test determined that 13% of the genetic distance found was explainable by isolation by distance. Similarly, the spatial autocorrelation analysis of the sequences of the individuals showed significant monotonic clines related to isolation by distance (however, when only haplotypes were used no spatial structure was detected). AIDA of the CR detected genetic patches with a diameter of around 30 km, and later isolation-by-distance. GLIA detected the sample from San Martin River (Iténez-Guaporé river basin) as the most differentiated of all the samples. The results are consistent with female philopatry. In other species of marine dolphins such as the bottlenose dolphins, patterns of population structure across very small distances are commonly described [153,154]. Other authors determined fine-scale population structure within the Adriatic Sea for bottlenose dolphins consistent with local reports of strong site fidelity [155,156]. The female site fidelity is observed elsewhere in European waters, such as the Moray Firth (Scotland), the Shannon Estuary (Ireland), and the Sado Estuary (Portugal). In all locations, small populations of females show strong site-fidelity to semi-enclosed bays, with limited interaction with populations outside the bays [157,158]. Many studies have demonstrated that sexual differences in dispersal trends exist [159,160,161] and that this affects the genetic structure of the species. For example, some studies [159,160,161] demonstrated that migrant sex is the vector of gene flow and a factor of global genetic homogeneity. Philopatric sex introduces internal homogeneity within the lineages (at least promoting gene correlation between individuals in the same lineage or close geographic lineages) and genetic heterogeneity among the most distant lineages. One study [5] used Chesser’s social models to determine asymptotic values of the F-statistics. It was showed that bufeos in the Peruvian Napo and Curaray rivers have a social reproductive system. Based on their findings, the basic genetic lineages could be composed of seven reproductive females per lineage in each breeding period (with high site fidelity) and the number of reproductive males per lineage is not important. Furthermore, on average, within each lineage there was a reproductive male with four females, and thus polygyny exists within this dolphin species (with some migrating males, although others presenting site fidelity). There was also a probability of 0.30 that females of the same lineage choose the same male for breeding. Although conditions could be different in the Bolivian rivers, these facts [5] agree with the genetic results in the current study, specifically regarding the philopatry of the bufeo females that was detected in spatial genetic analyses with mt markers.

In contrast, nuclear microsatellites did not detect isolation by distance and even the two first DCs, in the autocorrelation analyses, were significantly negative. This finding could agree with the fact that, while the females were philopatric, some males migrated, on average, more than 30 km to successfully breed. Therefore, the population dynamics of females and males were different. This was due to habitat preference and reflected in the spatial structure of the molecular markers. Male bufeos are primarily found in the main channels of large rivers, whereas females prefer more protected areas, such as flooded forests and its channels and lakes [142]. Several studies have determined the minimum home ranges of bufeos. These estimates varied from 12.1−27.9 km^2^ (Juruena River, Brazil) to 60.5−233.9 km^2^ (San Martín River, Bolivia). Some of the distances travelled by individual male bufeos are worth mentioning, such as (1) 7.5 km in the Juruena River, (2) 36.2 km in the Amazon River (Colombia and Perú), and (3) 51.8 km in the Orinoco River (Colombia and Venezuela). Some authors [162,163] reported 225 km for one bufeo in the Japurá River (Brazil). The distance travelled of four individual bufeos was also determined (mean of 61 km with a maximum of 220 km) [164]. Likewise, it was detected that there were movement distances ranging from 50 to 200 km for bufeos in the Cuyabeno and Lagartococha rivers (Ecuador) [165]. The longest total distance traveled by a specimen was recorded by a male in one of the Bolivian rivers where we sampled bufeos. It was 297.9 km [166]. In all cases, the maximum distances were carried out by males. Long fish migrations (bufeos predated more than 43 fish species, many of them with migratory patterns) from 500 km up to 3000 km [167] can potentially explain the relatively long-distance values recorded in tagged bufeos. It was shown, through satellite tracking, that the Amazon bufeo traveled between 20 and 100 km per day, whereas other dolphins, generally females, stayed for weeks in a small (1 km^2^) area [163]. Using photo identification, it was reported that the average range of movement detected for bufeos in Bolivia is at least 60 km, or 3 to 10 km daily [164]. River dolphins are highly mobile top predators that can cover long distances (hundreds of kilometers) in relatively short periods (days), with a possible differential use of habitat by males and females [162,163,166,168,169]. In fact, typical daily movements of up to 20 km, and individuals swimming at sustained speeds of 3–6 km/h were reported [170].

On the other hand, the sequences at the *DQB-1* locus did not show any spatial trend for any of the procedures used. This could mean that selective pathogen and parasite pressures are very similar throughout the Bolivian rivers, as well as for other Neotropical rivers (Perú and Colombia–Venezuela) where *DQB-1* alleles were shared. Henceforth, spatial structure analyses could help to determine the different evolutionary events which occur for different kinds of molecular markers.

### 4.4. Demographic Changes

The MJN procedure revealed that the diversification of the mtDNA haplotypes in the Bolivian rivers studied began around 180,000–130,000 YA (for CR). We know that selection can affect effective population size, reducing the effective number for a time and increasing the coalescence rate later, which could increase “artificially” the temporal splits among populations. The same occurs with small changes in the mutation rate (μ), which can greatly affect the effective number and, in turn, the estimated divergence time. For instance, if the estimated mutation rate is smaller than the real one, the temporal-divergence split will also be smaller than the real one for a determined effective size. Contrarily, if the estimated mutation rate is higher than the real one, this increases the temporal-divergence estimates in an unreal way. Despite this, our temporal mtDNA diversification process agrees quite well with that detected by another work [10]. They estimated isolation between the upstream and downstream groups of bufeos across the rapids of the Madeira River to have occurred around 122,000 YA (95% HPD 32,000–283,000 YA). They performed an IMA2 analysis using a harmonic mean of the average mammalian substitution rates. When they used the upper- and lower-bound substitution-rate values from another study [171], their estimated values ranged from a mean of 97,000 (95% HPD 25,000–226,000 YA) to 163,000 YA (95% HPD 43,000–379,000 YA). Henceforth, both studies concluded that at the beginning of the fourth great Pleistocene glaciation, the rapids of the Mamoré–Iténez (Guaporé) River basin and the Madeira River basin influenced the diversification of mitochondrial haplotypes in *I. boliviensis*. This divergence was not due to the formation of the upper Madeira River rapids, but rather, a different event. The formation of the rapids most likely resulted from the rise of the Fitzcarald Arch in the mid-to-late Pliocene, around 4 MYA [172,173]. The period of mtDNA haplotype differentiation, which occurred around 170,000–100.000 YA, coincided with beginning of the last large Pleistocene glacial period (transition from the Middle to the Late Pleistocene) [174]. This could be connected to the haplotype diversification detected in the bufeos of the Mamoré and Madeira rivers. For instance, although 27 possible major climatic changes with a periodicity of 100,000 years during the Pleistocene in northern South America were defined [175], there only is clear evidence for the last glaciation [176,177]. This last cold period began an estimated 116,000 YA, which is the same time of estimated divergence for many of the pairs of haplotypes of *I. boliviensis*. Probably, these cold periods substantially decreased the level of the waters in the rivers of the Mamoré–Iténez (Guaporé) and Madeira basins. Under conditions of very low waters, the rapids become a partial barrier to the dispersion of the dolphins within this geographical area and affected the hydrological dynamics of these rivers. Additionally, the reticulations observed in the MJN agrees well with this scenario. Some lineages of bufeos should be isolated in several riverine refuges when the levels of water decreased during the last phase of the Pleistocene and accumulated unique mutations, but when the waters rose again some lineages expanded and widely dispersed their genetic characteristics across the major fraction of the Mamoré River basin. For this reason, we found exclusive and private haplotypes, together with well-expanded haplotypes, in the same population.

All the statistics used to detect demographic changes with mtDNA detected significant population expansions, except for the mismatch distribution. Bolivian River dolphins showed considerably lower mtDNA diversity than did *Inia* populations from other basins (Amazon and Orinoco). Based on statistical analysis of the mtDNA marker, it seems clear that the Bolivian population crossed an initial bottleneck with posterior population expansion. BSP for CR detected a female population decrease which began around 60,000 YA, with a strong dip around 20,000 YA followed by an increase in the last 500 YA. Other marine animals showed a strong decline 20,000–12,000 YA (Last Glacial Maximum, LGM) with a subsequent population expansion. The killer whale population (*Orcinus orca*) declined during LGM and then expanded [178,179]. Similarly, the low diversity observed in loggerhead sea turtles in Eastern Mediterranean [41] has been linked with excessively low temperature for successful hatching during the LGM, which would imply that present loggerhead turtles are descendent from post-glacial colonizers. Additionally, it was suggested that differentiation of certain estuarine/bay populations of bottlenose dolphins in the North Atlantic has been achieved post LGM [180]. This suggests that the LGM might have had profound effects not only on the Mediterranean, North Atlantic, and Pacific marine fauna but also on the bufeos of the Bolivian rivers.

The nuclear microsatellite results were complementary to those obtained with the mtDNA. The Kimmel test [121] detected an initial bottleneck with posterior population increase. An interesting result from this analysis is that if the value is β > 1 (ln β > 0), then it will be present during several thousand generations (from 5000 to 10,000 generations) before these values show the signature of a population expansion (β < 1 (ln β < 0)). If, in the case of bufeos, a generation is about seven years, then the population expansion process after the initial bottleneck is not older than 35,000 Y (5000 generations; more probable) or 70,000 Y (10,000 generations; less probable). This could mean that more than one bottleneck has occurred throughout the natural history of *I. boliviensis*. Also, other tests [123,127] detected an intense bottleneck during the foundation of the population. The Zhivotovsky et al. test [123] is more powerful than the Kimmel test [121] for detecting strong bottlenecks in the initial formation of a population, or during its history. It has a greater potential for detecting when populations reach a demographic stability after a rapid population growth. The values of S_k_ and V for this test over time weakly depend on the rate of population increase and the final population size. Only extreme differences in the growth rate could produce substantial differences between these statistics. The Bolivian bufeos presented extremely different S_k_ and V estimates. The negative S_k_ value and the small V value demonstrated the existence of a strong bottleneck in this population. The question is whether this bottleneck occurred just before or during the population expansion. This last event may greatly influence S_k_ but only slightly affect V (Figure 6 from [123]). In the Bolivian bufeo population, both S_k_ and V were extremely affected. Therefore, the striking bottleneck was in the original Bolivian population. This result disagrees with previous hypotheses [181], which claimed that the original Bolivian population gave origin to all other bufeo populations in the Amazon and the Orinoco basins. An extreme bottlenecked population cannot generate other populations with more genetic diversity than itself. On the other hand, S_k_ is not affected by different mutation rates in the microsatellites (at least if this rate does not change over time), whereas V is greatly affected. Therefore, because all the microsatellites studied were dinucleotides, different S_k_ values could not be attributed to different mutation rates. Also, we have no evidence that the mutation rate changed, at least when the population began to grow. If the mutation rate had increased, then both S_k_ and V would have been higher than the values found. The k and g tests [129,130] yielded no evidence of significant population expansions for the Bolivian river dolphins. This showed that these tests have less power to detect population changes than do the previously quoted tests.

These demographic changes, together with the low levels of genetic diversity herein reported for the Bolivian bufeos, indicate that the current *I. boliviensis* originated as a small founder population split off from the central Amazon into the upper reaches of the proto-Madeira River. The population became isolated via a geomorphic event that prevented gene flow with other *Inia* populations for a considerable amount of time. This could have begun around 4–3 MYA [172,173]. This explains the low levels of gene diversity found in the three types of different molecular markers we used in this study. This event is likely associated with the migration of rivers draining the Andean front in response to eastward propagation of the Andean fold and thrust belt [182,183]. The modern upper Madeira River has migrated to the westernmost edge of the resistant Brazilian shield, which now constrains its course, as can be clearly seen by its unique NW–SE flow direction. The flow of the Madeira over the rocky western edge of the shield creates the barrier isolating *Bolivian bufeos* from the mainstem of the Amazon basin. This first event agrees with the estimated temporal splits between the ancestors of *I. geoffrensis* and *I. boliviensis* (2.6–6.2 MYA, [3]; 1.5–3.8 MYA, [9]; 3.1 MYA, [143]; and 2.87 MYA, [184]). The history of population changes after this founder event is recorded in the mitochondrial sequences and in the nuclear markers studied. Around 180,000–100,000 YA, at the beginning of the fourth great Pleistocene glaciation (and maybe with the influence of the Eemian interglacial period, around 150,000 YA) climatological changes (more than geological changes) decreased the depth of the rivers within the Madeira basin. With very low waters, the rapids became population barriers, and possibly helped to create different mt haplotypes, which later expanded throughout the Bolivian rivers. Once again, during the LGM, another population reduction occurred, followed by another small increase over the last few centuries. For this reason, different tests for mt and nu markers detected different population bottlenecks and expansions for *I. boliviensis*. These results invalidate the hypotheses proposed by Grabert [181] for the origin of *Inia* in South America. This hypothesis proposed a Middle Miocene Pacific entrance prior to the uplift of the northern Andes. A Pacific coastal Iniidae entered a large Bolivian lake (sub-Andean freshwater molasses) during the Miocene (15 MYA). *I. boliviensis* then later appeared in this molasse lake during the Pliocene, around 5 MYA. It became highly adapted to the turbidity of these waters. Sometime afterwards, approximately 1.8 MYA, *I. boliviensis* entered the Amazon basin through the Purus or the Iquitos gates, eventually causing the appearance of *I. g. geoffrensis*. Later, about 10,000 YA, *I. g. humboldtiana* formed from *I. g. geoffrensis* within the Cassiquiare channel. This hypothesis is totally rejected. Therefore, *Inia boliviensis* could not be the original form of *Inia*. Its smaller cerebral capacity and greater number of teeth are not considered ancestral characteristics in *Inia*. In contrast, they are derived characteristics, which could represent a violation of Willingston’ law. A strong gene drift and the prolonged bottleneck effects could change specific morphological characteristics following a genetic revolution [152] or a “flash-crash” process [185,186].

The bufeo is considered an ideal biological model in helping us to understand the dynamics of aquatic ecosystems because it has a small population size and high habitat requirements, and differentially uses its habitats. Moreover, bufeos are top predators, have long gestation periods, and demonstrate long-term parental care. Therefore, bufeos are good bioindicators of aquatic environments because they are sensitive to the environmental changes [187]. It was determined that there was a decrease of 35.7% in the number of bufeos in a population in the Orinoco (Colombia) over six years [168]. For the Colombian Amazon, it was reported that there was a reduction of 10% per year in a populations of *I. geoffrensis* between 1993 and 2007 [188]. These reductions can be explained by the increased number of bufeos captured and used as bait for the mota (*Calophysus macropterus*) fishery. Bufeos need environmental protection. Molecular-population genetic studies, like the current one, provide critical information in helping us to determine the genetic history and diversity of the bufeo. The use of different kinds of molecular markers is relevant, as we show in the current work, because some of them are considered to have a neutral behavior (mt CR, which is illustrative of the female colonization process, and nu microsatellites, which detected the important vector of gene flow which the males have), but others (*DQB-1* locus) clearly respond to pathogen selective pressure, which is similar in different Neotropical rivers. Such information is crucial to developing a meaningful conservation plan.

## Figures and Tables

**Figure 1 genes-15-01012-f001:**
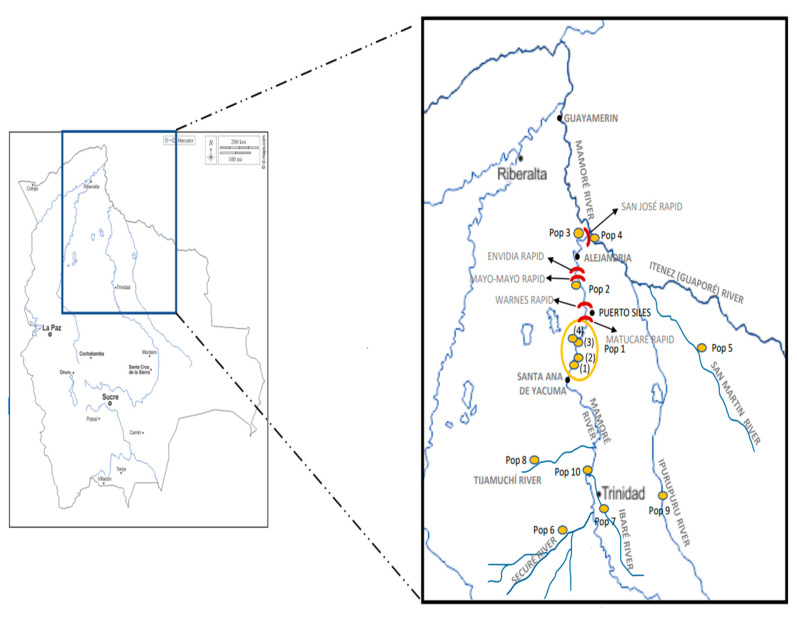
Map with the 10 *Inia boliviensis* populations sampled in the Mamoré and Iténez (Guaporé) rivers (Bolivia) for different molecular markers. Pop 1 = four sampling points (Porvenir, Bellaunión, Cerrito, and Iruyañez River mouth), all in the Mamoré River; Pop 2 = Bolivar (Mamoré River); Pop 3 = El Corte (Mamoré River); Pop 4 = El Azul (Iténez River); Pop 5 = San Martín River (Iténez River); Pop 6 = Securé River (Mamoré River); Pop 7 = Ibaré River (Mamoré River); Pop 8 = Tijamuchí River (Mamoré River); Pop 9 = Ipurupuru River (Iténez River); Pop 10 = mouth of Ibaré River with Mamoré River. The rapids in the Mamoré River are also shown on the map.

**Figure 2 genes-15-01012-f002:**
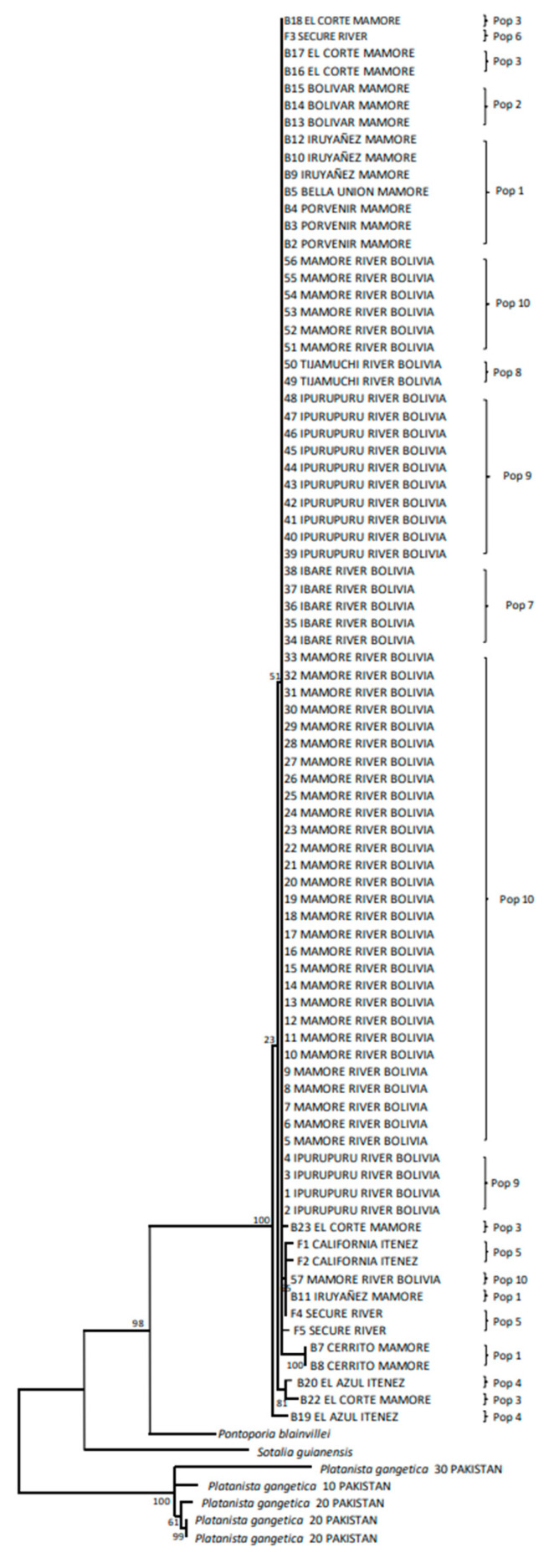
Maximum likelihood tree showing the relationships among 82 bufeos (*Inia boliviensis*) sampled in Bolivian rivers sequenced at the mt control region. The number in the nodes are the bootstrap percentages.

**Figure 3 genes-15-01012-f003:**
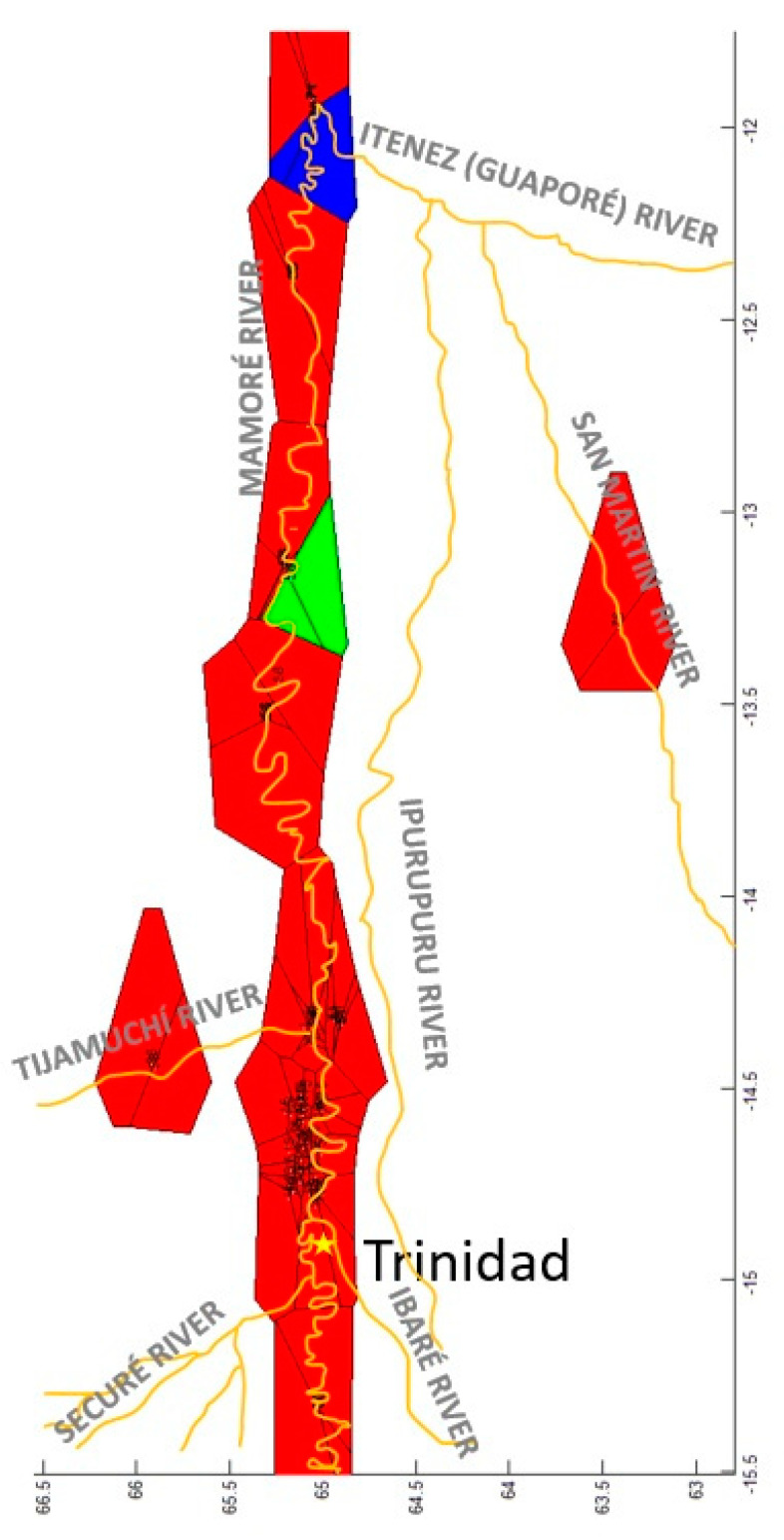
Different populations of bufeos (*I. boliviensis*) detected by means of BAPS in Bolivian rivers for the mt Control Region with 82 bufeo specimens; three populations were detected.

**Figure 4 genes-15-01012-f004:**
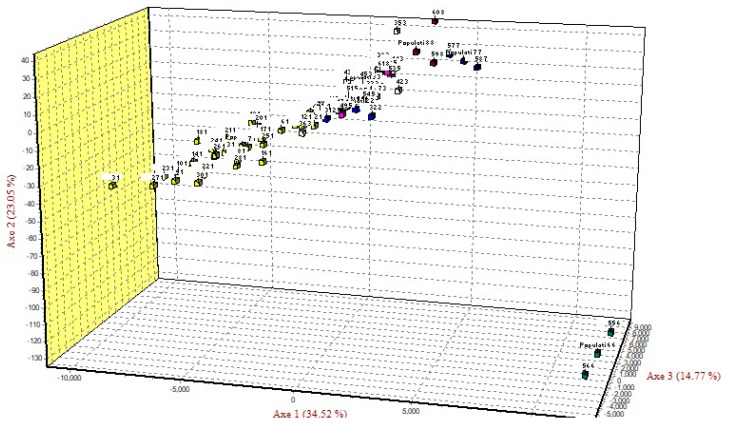
Correspondence factorial analysis showing the relationships of 61 bufeos (*I. boliviensis*) analyzed for 10 DNA microsatellites in Bolivian rivers. The specimens sampled in the Iténez (Guaporé) River were the most differentiated.

**Figure 5 genes-15-01012-f005:**
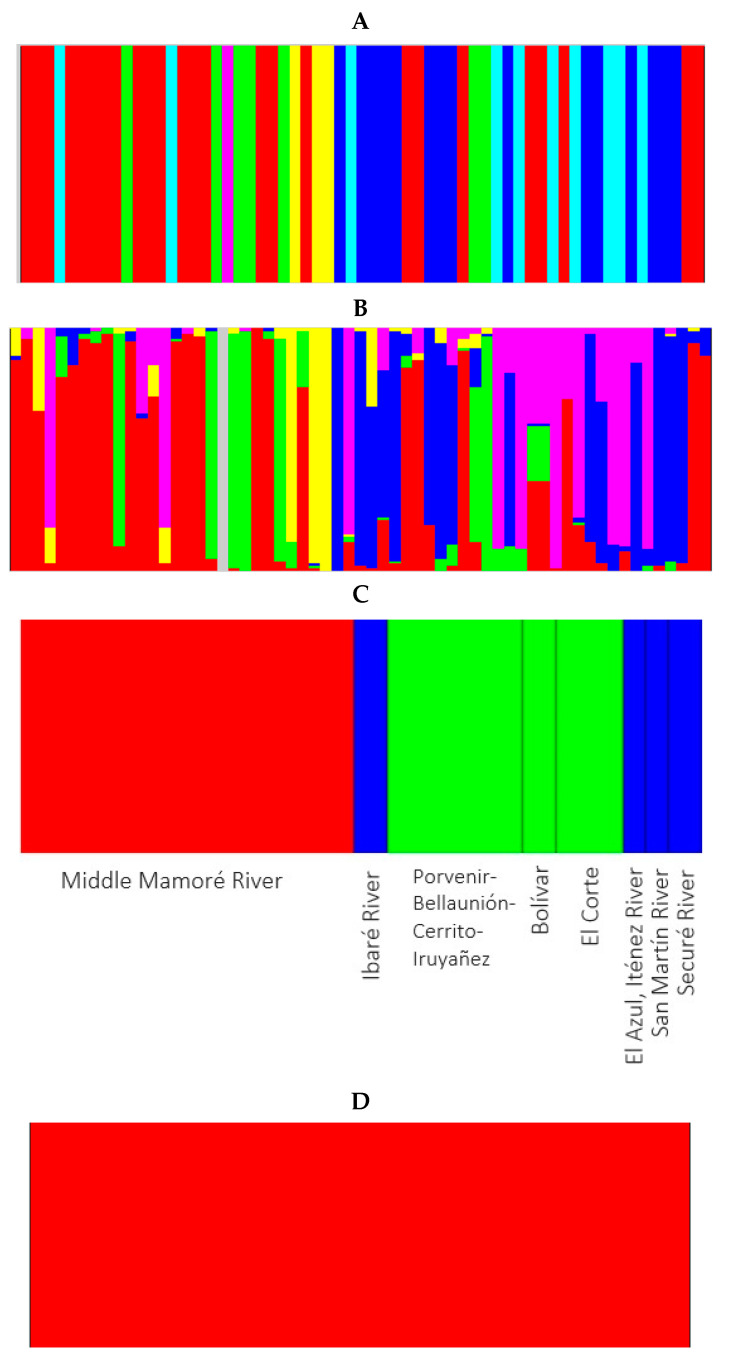
Different populations of bufeos (*Inia boliviensis*) detected by means of BAPS in Bolivian rivers using 10 DNA microsatellite markers. (**A**) Sixty-one specimens assuming no admixture among different localities; six populations were detected. (**B**) Sixty-one specimens assuming admixture among different localities; six populations were detected. (**C**) Eight populations assuming no admixture among them; three populations were detected. (**D**) Eight populations assuming admixture among them; one population was detected. Each different color indicates a different population.

**Figure 6 genes-15-01012-f006:**
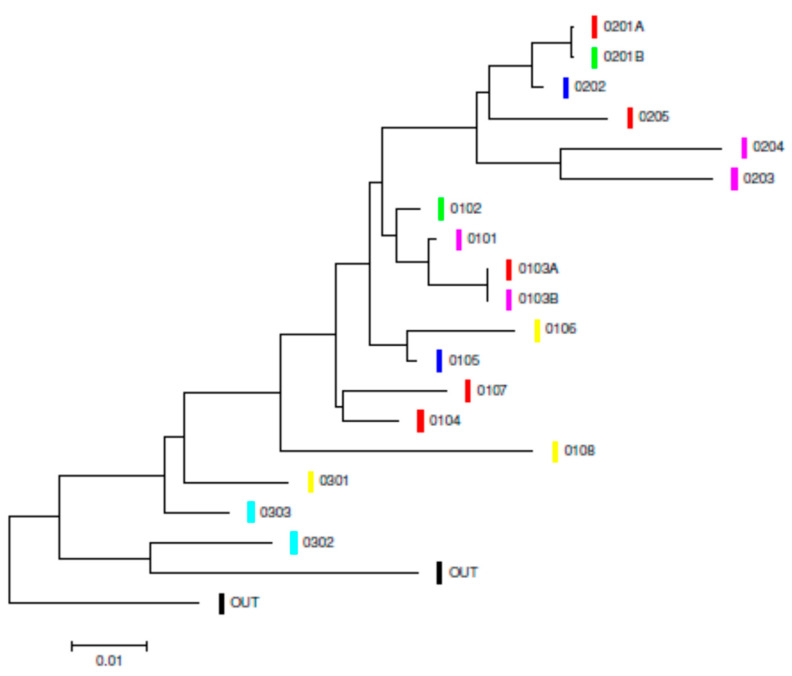
Minimum evolution tree with the Kumar tree analyzing *DQB-1* gene sequences in bufeos (*I. boliviensis*) sampled in Bolivian rivers. This three shows the relationship among the 11 *DQB-1* alleles found in the Bolivian rivers and those found in *Inia* populations from the Peruvian Amazon rivers and from the Orinoco (Colombian and Venezuelan) basin rivers.

**Figure 7 genes-15-01012-f007:**
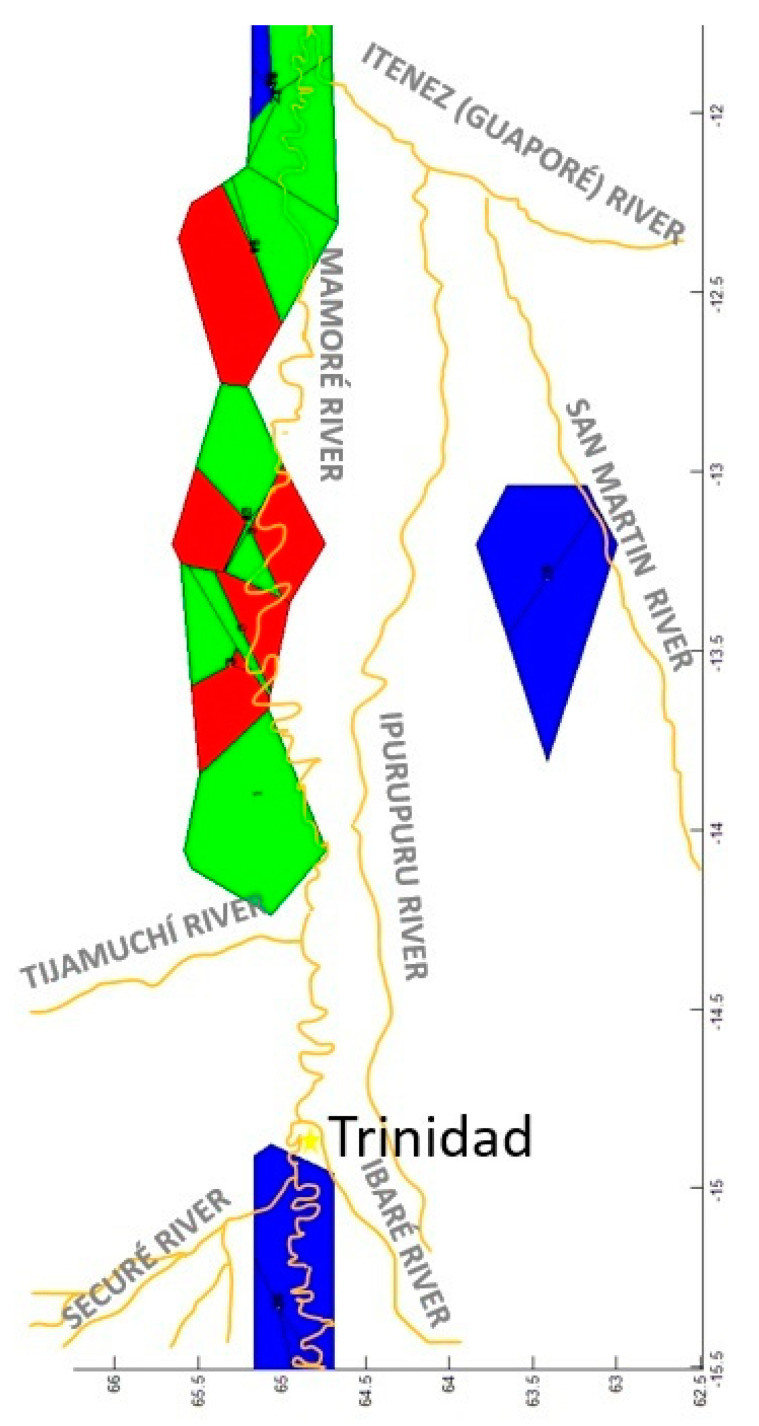
Different populations of bufeos (*I. boliviensis*) detected by means of BAPS in Bolivian rivers using *DQB-1* gene sequences. Twenty-three (23) specimens were analyzed, and four populations were detected.

**Figure 8 genes-15-01012-f008:**
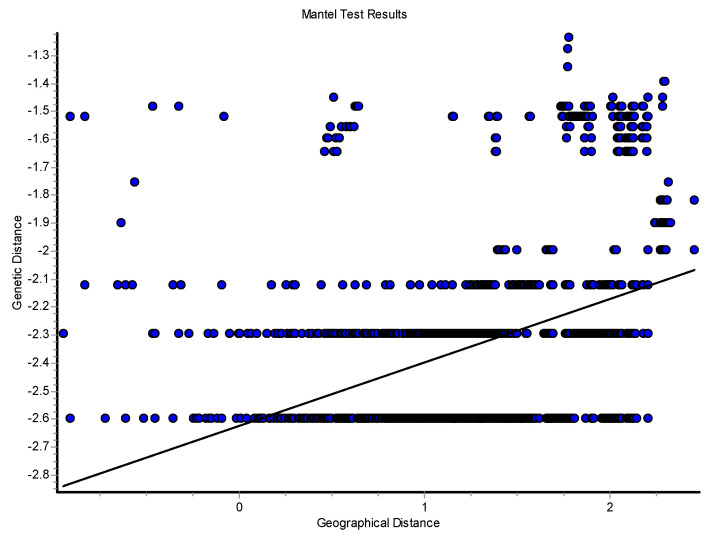
Mantel test between the geographic and genetic distances for specimens of bufeos (*I. boliviensis*) sampled in Bolivian rivers for 82 specimens at the mt control region. Significant spatial structure was found.

**Figure 9 genes-15-01012-f009:**
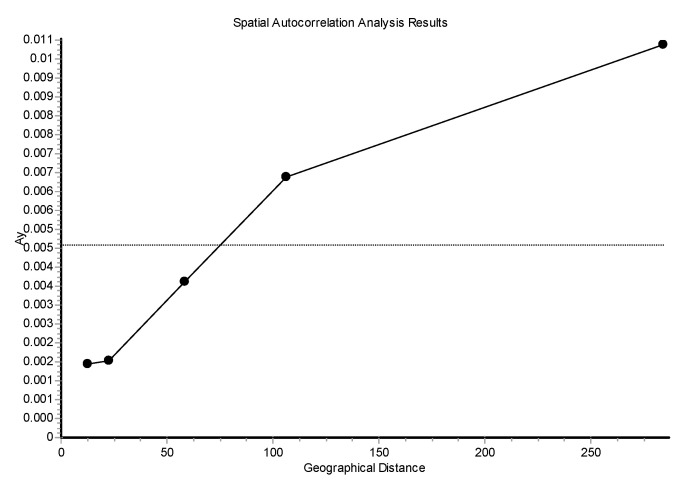
Correlogram with the Ay statistic and five distance classes from a spatial autocorrelation analysis for bufeos (*I. boliviensis*) in the Bolivian rivers for 82 specimens at the mt control region. Significant isolation by distance was detected.

**Figure 10 genes-15-01012-f010:**
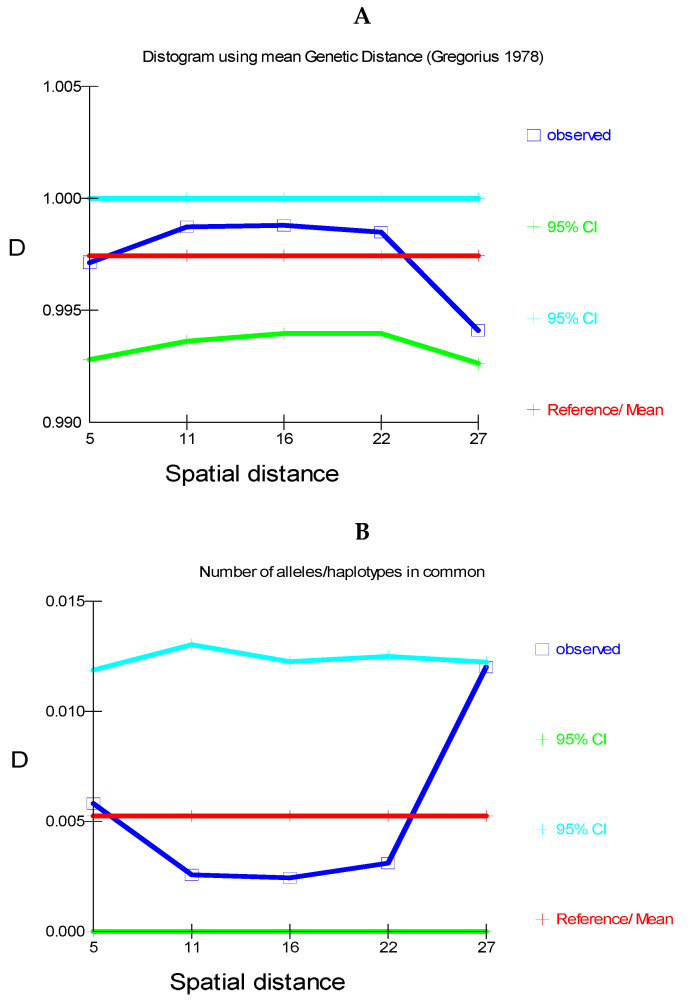
Distograms using five distance classes to analyze the spatial genetic structure of the bufeo (*I. boliviensis*) in the Bolivian rivers at the mt control region. (**A**) For the haplotypes found using the Gregorious distance [96]. (**B**) For the haplotypes found using haplotypes in common. No significant pattern was found with these procedures.

**Figure 11 genes-15-01012-f011:**
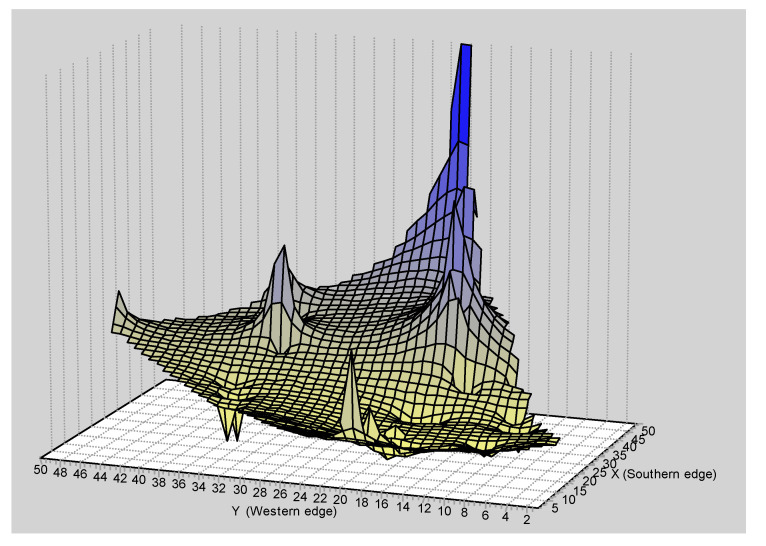
Genetic-landscape interpolation analysis (GLIA) for bufeos (*I. boliviensis*) across Bolivian rivers for 82 specimens analyzed at the mt control region. Isolation by distance and genetic patches were detected with this procedure. The color gradient shows the isolation by distance, meanwhile the peaks indicate some genetic patches.

**Figure 12 genes-15-01012-f012:**
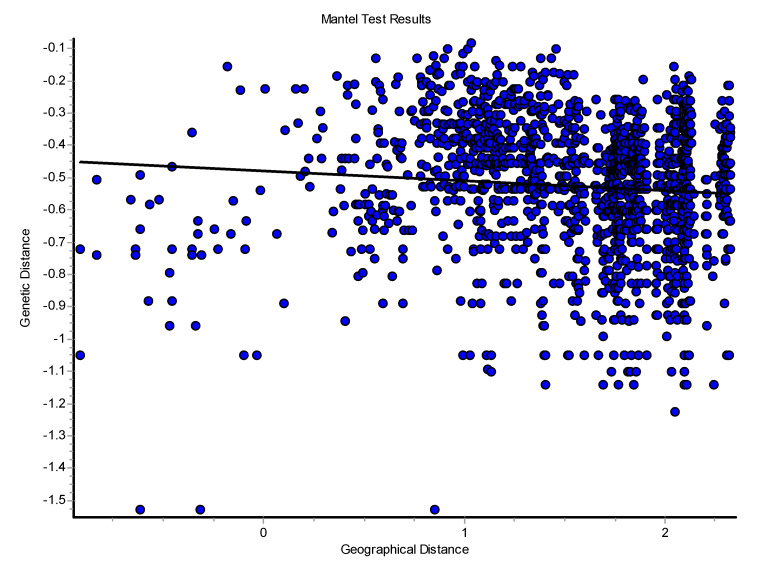
Mantel test between the geographic and genetic distances for specimens of bufeos (*I. boliviensis*) sampled in Bolivian rivers. Sixty-one (61) specimens analyzed at 10 DNA microsatellites. No spatial structure was found with DNA microsatellites.

**Figure 13 genes-15-01012-f013:**
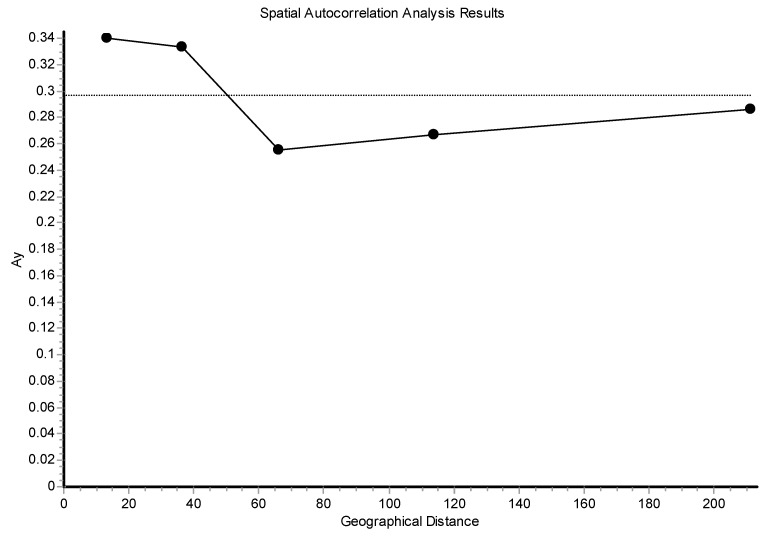
Correlogram with the Ay statistic and five distance classes from a spatial autocorrelation analysis for bufeos (*I. boliviensis*) in the Bolivian rivers at 10 DNA microsatellite markers. Sixty-one (61) specimens were analyzed for this procedure. No spatial structure was found with DNA microsatellites. In fact, the most nearby specimens were the most differentiated ones.

**Figure 14 genes-15-01012-f014:**
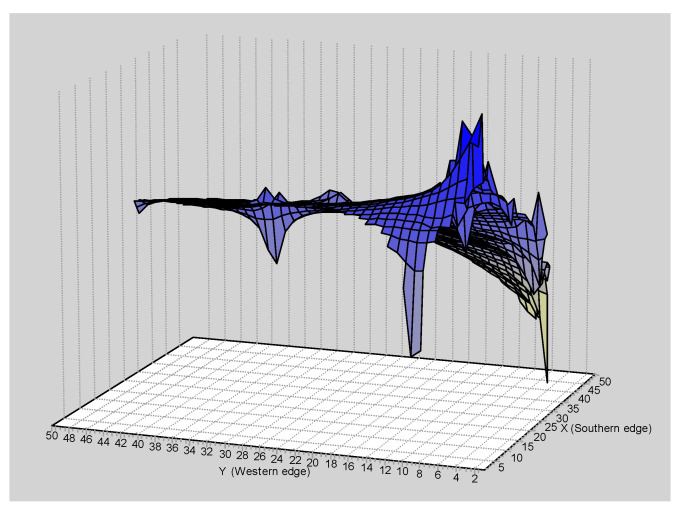
Genetic-landscape interpolation analysis (GLIA) for bufeos (*I. boliviensis*) across Bolivian rivers at 10 DNA microsatellite markers. Sixty-one (61) specimens analyzed for this procedure. No clear genetic patches were detected with this technique.

**Figure 15 genes-15-01012-f015:**
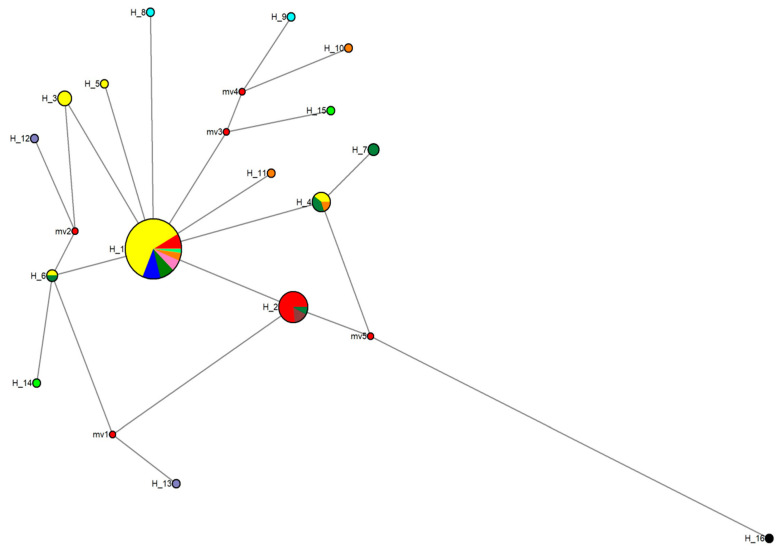
Median-joining network (MJN) with haplotypes identified in 82 specimens of bufeos (*I. boliviensis*) sampled in Bolivian rivers at the mt control region. Green = Pop 1; pink = Pop 2; orange = Pop 3; light blue = Pop 4; lilac = Pop 5; light green = Pop 6; dark blue = Pop 7; brown = Pop 8; red = Pop 9; yellow = Pop 10; black = *Pontoporia blainvillei* (outgroup). Small red circles indicate missing intermediate haplotypes.

**Figure 16 genes-15-01012-f016:**
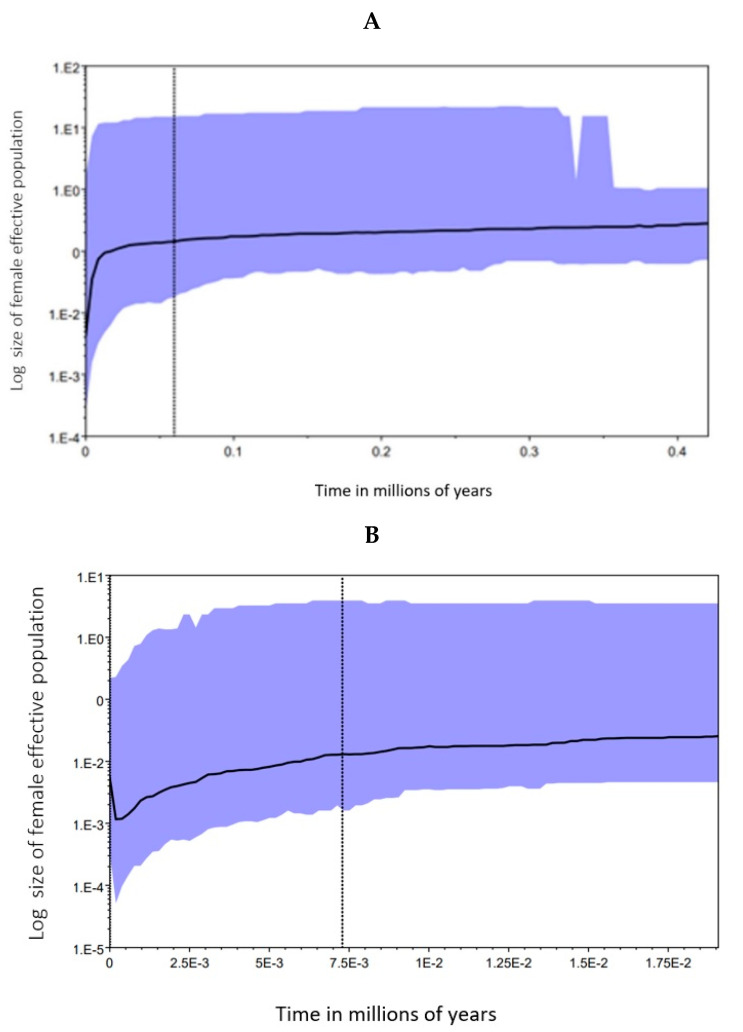
Bayesian skyline plot analyses (BSP) to determine possible demographic changes across the natural history of bufeos (*I. boliviensis*) in Bolivian rivers with (**A**) mt control region in the last 400,000 years and (**B**) with mt control region in the last 17,500 years. Population declination was observed during the last phase of the Pleistocene and a light population increase was detected in the recent centuries.

**Table 1 genes-15-01012-t001:** Overall genetic heterogeneity and indirect gene flow (Nm) statistics among all the *I. boliviensis* populations considered at the mitochondrial control region (CR) in Bolivian rivers. * Significant probability (*p* < 0.001); df = degree of freedom, Nm = gene flow.

Genetic Differentiation Estimated	*p*	Gene Flow	
χ^2^ = 330.4df = 126	0.00001 *		
H_ST_ = 0.3005	0.00001 *	γ_ST_ = 0.3771	Nm = 0.83
K_ST_ = 0.2034	0.00001 *	N_ST_ = 0.2504	Nm = 1.50
K_ST_ * = 0.3913	0.00001 *	F_ST_ = 0.2509	Nm = 1.49
Z_S_ = 1273.5639	0.00001 *		
Z_S_ * = 6.9527	0.00001 *		
S_nn_ = 0.3989	0.00001 *		

**Table 2 genes-15-01012-t002:** Genetic differentiation by means of exact probability tests with Markov chains (below diagonal) and gene flow estimates (above diagonal) of *I. boliviensis* population pairs sampled in different Bolivian rivers analyzed for the mitochondrial control region (CR). 1 = Middle Mamoré River population (Porvenir, Bellaunión, Cerrito, mouth of Iruyañez River with Mamoré River), *n* = 10; 2 = Bolivar population (Mamoré River), *n* = 3; 3 = El Corte population (Mamoré River), *n* = 5; 4 = El Azul population (Iténez River), *n* = 2; 5 = San Martín River population (Iténez River), *n* = 2; 6 = Securé River (Mamoré River), *n* = 3; 7 = Upper Ibaré River (Mamoré River), *n* = 5; 8 = Tijamuchí River (Mamoré River), *n* = 2; 9 = Ipurupuru River (Iténez River), *n* = *14*; 10 = Mouth of Ibaré River with Mamoré River (Mamoré River), *n* = 36. * *p* < 0.05; ** *p* < 0.0001, significant heterogeneity, *n* = sample sizes.

Population	1	2	3	4	5	6	7	8	9	10
1		infinite	10.065	0.560	1.671	36.489	13.648	12.024	1.365	1.312
2	0.713 ± 0.005		Infinite	0.434	0.600	infinite	Infinite	0.000	0.439	infinite
3	0.708 ± 0.004	1.000 ± 0.000		1.417	2.342	infinite	Infinite	32.500	0.953	1.061
4	0.121 ± 0.005	0.093 ± 0.002	0.521 ± 0.006		1.062	0.758	0.221	0.687	0.101	0.060
5	0.115 ± 0.004	0.098 ± 0.003	0.537 ± 0.004	1.000 ± 0.000		2.684	0.288	1.500	0.203	0.119
6	0.332 ± 0.004	0.403 ± 0.002	1.000 ± 0.000	1.000 ± 0.000	1.000 ± 0.000		2.143	1.263	0.457	0.720
7	0.404 ± 0.003	1.000 ± 0.000	0.168 ± 0.003	0.046 ± 0.001 *	0.045 ± 0.002 *	0.108 ± 0.003		0.000	0.362	infinite
8	0.184 ± 0.004	0.103 ± 0.002	0.236 ± 0.003	0.335 ± 0.002	0.329 ± 0.002	0.400 ± 0.006	0.045 ± 0.001 *		infinite	0.199
9	0.005 ± 0.001 *	0.050 ± 0.001 *	0.006 ± 0.001 *	0.008 ± 0.000 *	0.009 ± 0.001 *	0.007 ± 0.001 *	0.010 ± 0.000 *	1.000 ± 0.000		0.392
10	0.008 ± 0.001 *	1.000 ± 0.000	0.048 ± 0.003 *	0.011 ± 0.002 *	0.008 ± 0.001 *	0.036 ± 0.003 *	1.000 ± 0.000	0.005 ± 0.000 *	0.000 ± 0.000 **	

**Table 3 genes-15-01012-t003:** Net Kimura 2P genetic distances in percentages (%) among all the population pairs of *I. boliviensis* sequenced at the mitochondrial Control Region (CR) in diverse Bolivian rivers and standard deviations in percentages (%). 1 = Middle Mamoré River population (Porvenir, Bellaunión, Cerrito, mouth of Iruyañez River with Mamoré River), *n* = 10; 2 = Bolivar population (Mamoré River), *n* = 3; 3 = El Corte population (Mamoré River), *n* = 5; 4 = El Azul population, (Iténez River), *n* = 2; 5 = San Martín River population (Iténez River), *n* = 2; 6 = Securé River (Mamoré River), *n* = 3; 7 = Upper Ibaré River (Mamoré River), *n* = 5; 8 = Tijamuchí River (Mamoré River), *n* = 2; 9 = Ipurupuru River (Iténez River), *n* = 14; 10 = Mouth of Ibaré River with Mamoré River (Mamoré River), *n* = 36; 11 = *Pontoporia blainvillei*; 12 = *Sotalia guianensis*; 13 = *Platanista gangetica*; *n* = sample sizes.

Populations	1	2	3	4	5	6	7	8	9	10	11	12	13
1													
2	0.6 ± 0.2												
3	1.4 ± 0.3	0.7 ± 0.2											
4	3.2 ± 0.8	2.6 ± 0.7	2.8 ± 0.7										
5	1.5 ± 0.5	0.9 ± 0.5	1.6 ± 0.5	3.5 ± 0.9									
6	0.9 ± 0.2	0.3 ± 0.2	1.0 ± 0.2	2.8 ± 0.8	1.0 ± 0.4								
7	0.6 ± 0.2	0.0 ± 0.0	0.7 ± 0.2	2.6 ± 0.7	0.9 ± 0.5	0.3 ± 0.2							
8	0.6 ± 0.2	0.0 ± 0.0	0.7 ± 0.2	2.6 ± 0.7	0.9 ± 0.5	0.3 ± 0.2	0.0 ± 0.0						
9	0.6 ± 0.2	0.0 ± 0.0	0.7 ± 0.2	2.6 ± 0.7	0.9 ± 0.5	0.3 ± 0.2	0.0 ± 0.0	0.0 ± 0.0					
10	0.6 ± 0.2	0.0 ± 0.0	0.7 ± 0.2	2.6 ± 0.7	0.9 ± 0.4	0.3 ± 0.2	0.0 ± 0.0	0.0 ± 0.0	0.0 ± 0.0				
11	30.5 ± 3.1	30.0 ± 3.1	30.4 ± 3.1	31.4 ± 3.2	31.0 ± 3.1	30.5 ± 3.1	30.0 ± 3.1	30.0 ± 3.1	30.0 ± 3.1	30.1 ± 3.1			
12	50.3 ± 4.6	49.9 ± 4.6	50.5 ± 4.6	52.6 ± 4.8	49.6 ± 4.5	49.7 ± 4.5	49.9 ± 4.6	49.9 ± 4.6	49.9 ± 4.6	49.9 ± 4.6	46.8 ± 4.8		
13	57.9 ± 5.7	57.2 ± 5.7	57.8 ± 5.7	60.0 ± 5.8	57.8 ± 5.6	57.4 ± 5.6	57.2 ± 5.7	57.2 ± 5.7	57.2 ± 5.7	57.2 ± 5.7	57.7 ± 5.0	60.4 ± 5.7	

**Table 4 genes-15-01012-t004:** Analysis of molecular variance (AMOVA) for the mitochondrial control region (CR). * *p* < 0.05, ** *p* < 0.001. In bold, the best results.

Geographical Clusters	% Variance among Groups	% Variance within Populations
**2 groups**	**72.41% ****	**20.62% ****
3 groups	65.48% *	27.37% **
4 groups	38.56%	51.23% **
5 groups	Around 0%	67.94% **
6 groups	32.71%	55.49% **
7 groups	27.27%	59.30% **

**Table 5 genes-15-01012-t005:** Genotypic differentiation (below main diagonal) and genic differentiation (upper main diagonal) with exact G tests among *I. boliviensis* population pairs (eight populations) sampled in Bolivian rivers for 10 nuclear microsatellite markers. 1 = Middle Mamoré River population (Porvenir, Bellaunión, Cerrito, mouth of Iruyañez River with Mamoré River), *n* = 12; 2 = Bolivar population (Mamoré River), *n* = 3; 3 = El Corte population (Mamoré River), *n* = 6; 4 = El Azul population (Iténez River), *n* = 2; 5 = San Martín River population (Iténez River), *n* = 2; 6 = Securé River (Mamoré River), *n* = 3; 7 = Upper Ibaré River (Mamoré River), *n* = 3; 8 = Mouth of Ibaré River with Mamoré River (Mamoré River), *n* = 30; * *p* < 0.05; ** *p* < 0.001; *n* = sample sizes.

Population Pairs	1	2	3	4	5	6	7	8
1		0.9808	0.1499	0.0978	0.4787	0.5674	0.6324	0.0010 **
2	0.9869		0.6750	0.2712	0.0672	0.1923	0.2408	0.3624
3	0.3142	0.6908		0.0447 *	0.0094 *	0.0555	0.0975	0.00005 **
4	0.6932	0.5617	0.3706		0.3157	0.1647	0.1349	0.00001 **
5	0.6149	0.2347	0.0537	0.8844		0.2756	0.3122	0.0048 *
6	0.7501	0.2282	0.0980	0.3219	0.2960		0.9991	0.0189 *
7	0.6738	0.7128	0.3626	0.6721	0.6619	1.0000		0.4314
8	0.0051 *	0.4075	0.0008 **	0.1348	0.0396 *	0.0355 *	0.8117	

**Table 6 genes-15-01012-t006:** Nei’s (1978) genetic distance (in percentage, %) among *I. boliviensis* population pairs (eight populations) sampled in Bolivian rivers for 10 nuclear microsatellite markers. 1 = Middle Mamoré River population (Porvenir, Bellaunión, Cerrito, mouth of Iruyañez River with Mamoré River), *n* = 12; 2 = Bolivar population (Mamoré River), *n* = 3; 3 = El Corte population (Mamoré River), *n* = 6; 4 = El Azul population (Iténez River), *n* = 2; 5 = San Martín River population (Iténez River), *n* = 2; 6 = Securé River (Mamoré River), *n* = 3; 7 = Upper Ibaré River (Mamoré River), *n* = 3; 8 = Mouth of Ibaré River with Mamoré River (Mamoré River), *n* = 30; *n* = sample sizes.

Population Pairs	1	2	3	4	5	6	7	8
1								
2	0.3							
3	3.5	0.0						
4	11.4	8.5	7.2					
5	7.7	15.8	11.4	17.0				
6	4.2	7.4	4.3	12.7	6.3			
7	4.9	11.8	6.7	16.0	5.8	0.0		
8	4.8	5.3	8.9	18.8	18.3	9.7	6.5	

**Table 7 genes-15-01012-t007:** F_ST_ statistics (below main diagonal) and gene flow (upper main diagonal) among *I. boliviensis* population pairs (eight populations) sampled in Bolivian rivers for 10 nuclear microsatellite markers. 1 = Middle Mamoré River population (Porvenir, Bellaunión, Cerrito, mouth of Iruyañez River with Mamoré River), *n* = 12; 2 = Bolivar population (Mamoré River), *n* = 3; 3 = El Corte population (Mamoré River), *n* = 6; 4 = El Azul population (Iténez River), *n* = 2; 5 = San Martín River population (Iténez River), *n* = 2; 6 = Securé River (Mamoré River), *n* = 3; 7 = Upper Ibaré River (Mamoré River), *n* = 3; 8 = Mouth of Ibaré River with Mamoré River (Mamoré River), *n* = 30; * *p* < 0.05; ** *p* < 0.001; *n* = sample sizes.

Population Pairs	1	2	3	4	5	6	7	8
1		infinite	4.61	2.05	4.32	6.13	4.38	3.53
2	0.000		56.17	0.90	0.56	0.97	0.90	2.64
3	0.051	0.004		1.24	0.88	1.88	2.05	1.44
4	0.109	0.218 **	0.168 *		0.73	0.71	0.78	0.82
5	0.055	0.310 **	0.221 **	0.254 **		1.57	4.03	0.85
6	0.039	0.205 **	0.117 *	0.260 **	0.137 *		infinite	1.46
7	0.054	0.217 **	0.109 *	0.243 **	0.058	0.000		2.92
8	0.066	0.086	0.148 *	0.233 **	0.228 **	0.147 *	0.079	

## Data Availability

The data sets generated and analyzed during the current study are available from the corresponding author on reasonable request at the e-mails, mruizgar@yahoo.es, and mruiz@javeriana.edu.co. The GenBank accession numbers of the Bolivian river dolphins are AF52105-AF52126, KY296923-KY296947.

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
