# Peer review of "Are There Barriers Separating the Pink River Dolphin Populations (Inia boliviensis, Iniidae, Cetacea) within the Mamoré–Iténez River Basins (Bolivia)? An Analysis of Its Genetic Structure by Means of Mitochondrial and Nuclear DNA Markers"

_genes, 2024, doi:10.3390/genes15081012_

Round 1

Reviewer 1 Report

Comments and Suggestions for Authors

I must sincerely congratulate the authors. It is an excellent work, absolutely well constructed and carried out by testing numerous hypotheses, complete in every way. The statistical analyzes are excellently carried out. The content is quite important and sheds light on a problem relating to the various populations of Inia geoffrensis, debated and investigated several times. It is demonstrated that the system of rapids that would determine the supposed isolation of Inia boliviensis from Inia geoffrensis would not influence the genetic exclusivity of the population of the Bolivian population. Certainly its particularity, which in any case is also highlighted by some indisputable morphological and morphometric peculiarities, undoubtedly justify its subspecific status or Inia geoffrensis boliviensis but do not support the proposed specific status, in particular if one follows the Biological Concept of species, linked strictly to reproductive isolation. Many fine and detailed explanations are included in the very thought-provoking and quite definitive discussion.

Some individual points to change:

lines 69-70. Platanista gangetica is present in Bangladesh (Brahmaputra and Megna river systems)

lines 70-71. Platanista minor is exclusively from Indo basin (Pakistan) but not in Bangladesh!

Fig. 1 It needs to be redrawn more clearly. Place the map in the more general context of South America and the Amazon, highlighting the borders of the various countries.

Author Response

Referee 1

Referee 1 wrote:

I must sincerely congratulate the authors. It is an excellent work, absolutely well constructed and carried out by testing numerous hypotheses, complete in every way. The statistical analyzes are excellently carried out. The content is quite important and sheds light on a problem relating to the various populations of Inia geoffrensis, debated and investigated several times. It is demonstrated that the system of rapids that would determine the supposed isolation of Inia boliviensis from Inia geoffrensis would not influence the genetic exclusivity of the population of the Bolivian population. Certainly its particularity, which in any case is also highlighted by some indisputable morphological and morphometric peculiarities, undoubtedly justify its subspecific status or Inia geoffrensis boliviensis but do not support the proposed specific status, in particular if one follows the Biological Concept of species, linked strictly to reproductive isolation. Many fine and detailed explanations are included in the very thought-provoking and quite definitive discussion.

Answer:

We thanked and appreciated so much the general comments of Referee 1

Referee 1 said:

Some individual points to change:

lines 69-70. Platanista gangetica is present in Bangladesh (Brahmaputra and Megna river systems)

Answer:

It was corrected following the suggestion of Referee 1

lines 70-71. Platanista minor is exclusively from Indo basin (Pakistan) but not in Bangladesh!

Answer:

It was corrected following the suggestion of Referee 1

Fig. 1 It needs to be redrawn more clearly. Place the map in the more general context of South America and the Amazon, highlighting the borders of the various countries.

Answer:

We changed the map (Figure 1) introducing a map of South America and the countries in or near to the Amazon Basin following the suggestion of the Referee 1.

Thanks so much for all your comments and suggestions.

Prof. Manuel Ruiz-García. PhD

Reviewer 2 Report

Comments and Suggestions for Authors

I found this manuscript very accurate and well-written, even if could be wordy in some parts. Samples were sufficient and using a multiple marker approach gives this study good resonance, highlighted by the key results obtained.

I suggest the authors try to synthesize some periods during their re-writing linked to the very high similarity detected by the systems, which needs to be decreased by about 20% before the publication of this document.

Despite a running title, I suggest the authors shorten the current title, using one of the two halves which compose it. In the same way, try to avoid using words already reported in the title among keywords, substituting them with other related ones.

The abstract section may exceed the characters limit of the journal, so try to synthesize the results and introduction part to remain within the limit.

Figures 4, 12, 13, and 16a need to be improved in quality.

Best regards

Author Response

Referee 2

Referee 2 wrote:

I found this manuscript very accurate and well-written, even if could be wordy in some parts. Samples were sufficient and using a multiple marker approach gives this study good resonance, highlighted by the key results obtained.

I suggest the authors try to synthesize some periods during their re-writing linked to the very high similarity detected by the systems, which needs to be decreased by about 20% before the publication of this document.

Answer:

We thanked and appreciated so much the general comments from Referee 2. We have tried to shorten some parts of the manuscript, especially the Introduction and the bibliography

Referee 2 said:

1-Despite a running title, I suggest the authors shorten the current title, using one of the two halves which compose it. In the same way, try to avoid using words already reported in the title among keywords, substituting them with other related ones.

Answer:

We have reduced the title in some words, and we avoided repeat words in the title and the section of keywords

2- The abstract section may exceed the characters limit of the journal, so try to synthesize the results and introduction part to remain within the limit.

Answer:

We have shortened the Abstract following the suggestion from Referee 2

3- Figures 4, 12, 13, and 16a need to be improved in quality.

Answer:

We now send these figures (including also Figures 8 and 9) with the better resolution that we can.

Thanks so much for all your comments and suggestions.

Prof. Manuel Ruiz-García. PhD

Reviewer 3 Report

Comments and Suggestions for Authors

The manuscript is very thorough and detailed and many analyses and results are presented.  I think my main concern is about the sampling of individuals that is very meager for some collection localities relative to others, and the dataset may be overanalyzed given its small size (by modern standards, wherein genome-scale data can be quickly/cheaply collected and analyzed).  The authors might consider the following in revising/improving their manuscript.

1)   Line 2.  The title is among the longer ones I have seen for a journal article.  Most simply, could delete “(Control Region, Microsatellites, and

DQB-1 Gene Sequences)”?  This level of detail is not necessary as is in the abstract.

2)   Lines 64-132.  This paragraph is super-long and has a lots of details that may not be necessary.  Cutting some of the very specific details might improve readability since readers can go to the papers cited if want these additional details?  I am thinking maybe reducing length of this paragraph by 50% would better balance the other paragraphs in the introduction.  The study is focused on population structure and genetics, while this long paragraph is related to population threats to the species and some of this could be deleted or moved to the discussion if it ends up being relevant.

3)   Line 170.  I would contend that this paragraph has way too much detail and does not really relate to any materials and methods – could cut much of the details here as I do not see how they relate to materials and methods and would increase readability to trim this back.

4)   Line 182.  Where the authors say, “54 bufeos were sampled in four different areas (we named them “populations”)”, it might be better to call these ‘sampling sites’ as ‘population’ has a specific (although in my opinion vague…) definition, according to most textbooks.  The sampled sites might represent populations, but at the onset of the study, this is not known, so a more neutral term might be better?

5)   Line 200.  In some cases, it seems that very few specimens were sampled for particular ‘populations’.  Is this sampling adequate to determine evolutionary patterns in a robust way?  It seems that many alleles will be missed given this limited sampling of individuals?  However, I realize that sampling more specimens of this rare species may not be easy to achieve.

6)   Line 209.  Maybe state at the beginning of this paragraph that valid/approved animal protocols were used in the sampling that comply with Bolivian rules/regulations if such rules/regulations are relevant?

7)   Line 210.  I do not know the latest ‘rules’, but is it appropriate to describe the scientific helpers as ‘Indian fisherman’ or would the fisherman (or perhaps many readers be offended by that wording?  Maybe say ‘native Bolivian fisherman’, ‘indigenous Bolivian fisherman’, or some other?

8)   Line 221.  For the mt region, it would be helpful to indicate approximately how big the amplicons are to give the reader a feel for how many basepairs of the control region were sequenced?

9)   Line 253.  It is not clear from the methods to this point why the sampling was so different for the various molecular markers (e.g., only 23 individuals were sampled for the nuclear gene, many more individuals sampled for mtDNA).

10)  Line 295 and following paragraph.  As noted above, are there enough individuals sampled per ‘population’ to have enough power to do all of these various analyses.  Some of the ‘populations’ included very few individuals, and the a priori grouping into ‘populations’ might be considered somewhat arbitrary (that is, another researcher might have divided sampling localities into more ‘populations’ or merged ‘populations’ into a total lesser number of ‘populations’ from the start)?  More explanation in the methods on how decided on the various ‘populations’ might clarify to the reader exactly why specimens were divided into ‘populations’ the way they were by the authors?

11)   Line 312.  When did ‘global’ analyses, is it an issue that there is extensive missing data for some samples (e.g., the nuclear gene sampled from few individuals)?  Do the analyses/statistics used correct for/account for the missing data in an adequate way, or is this problematic?  Presumably the simulations deal with the missing data in terms of accounting for the missing data, but a concern is that with such small sample sizes, in particular for the nuclear locus (but also for the other markers), determining whether ‘populations’ are differentiated may be challenging given such small sample sizes?  If this is an issue (i.e., small sample sizes for various populations) that might impact interpretations, the authors should at the least comment on this issue in some way to put the reader at ease?

12)  Line 360.  These analyses do not assume a priori ‘populations’ which is good, but is it still a concern that sampling of individuals at some collection sites is so sparse?  Or, do many published studies sample so few specimens per collection site?  Again, the randomizations will ‘work’ to account for the low sampling at many sites, but I fear what patterns might be missed by sampling so few individuals per site.  Many analyses were done in this manuscript, but a concern is that a small sample of data has been ‘over-analyzed’ given the amount of data collected.

13)   Line 397.  Given the importance of this calibration method for the ‘molecular clock’ used (a particular evolutionary rate that is assumed), the authors should note here which taxa were used as a basis for the Inia molecular clock.  A reference is given, but the authors should note here whether the evolutionary rate for the D-Loop is based on this same region being used here and whether the rate is based on cetaceans (which generally have a low rate of molecular evolution among mammals) or some other taxa.

14) Line 410.  Is this evolutionary rate assumed here the same or different from the one noted above?

15)  Line 425 and line 437.  Does the ‘Coalescent Constant Population’ model’ assume neutrality and complete absence of selective sweeps?  If not, what is assumed?  If so, is it good to assume there are no selective sweeps, or can selective sweeps be differentiated from population bottlenecks, etc.?  Maybe a bit more here about assumptions of the methods is warranted, although some stuff on selection is mentioned here.

16)  Line 438.  In addition to changes in mutation rate, the mutation rate assumed, if completely wrong, will make it so that any correlation of changes at the molecular level with biogeographic or bioclimatics (etc.) factors at certain geologic times will potentially be way off.  So, just assuming one particular mutation rate is perhaps not the way to go.  Surely the uncertainty on this molecular rate of control region evolution in Inia is very high, and it would be good to incorporate this broad uncertainty into all analyses/interpretations that center on a molecular clock.  If a strict molecular clock is assumed, the data should be tested to see if the data fit a strict molecular clock or not.

17)  Line 505.  I am not personally familiar with the wide variety of microsatellite analyses done, so I leave comments on these analyses to other reviewers who have experience running microsatellites and analyzing them.

18)  Line 522.  I did not understand the basis for the statement, “These mitochondrial genetic diversity levels are medium for the CR.”  There should be some sort of citation(s) here to explain what ‘medium’ is.  Is this ‘medium’ for a cetacean, for a mammal, for a vertebrate, for vertebrates and invertebrates?

19) Line 548.  For the nuclear locus, a concern may be that the limited number of specimens analyzed did not allow a rigorous test of whether deviations from HW equilibrium is the case?

20)  Line 561.  The divergence times of these different genera of ‘river dolphins’ have been estimated many times and shown not to be particularly closely related, so this is either a point that is not worth making or can say that this result is consistent with basically all previous molecular clock analyses of cetaceans that have been done properly (e.g., various McGowen et al. papers).

21) Line 579.  In this table, I think it would be useful to show explicitly how many individuals were sampled for each of the 10 ‘populations’ shown since some ‘populations’ had many individuals and others had very few.  This would help readers interpret the results better maybe?

22)  Line 599 and other areas.  When refer to ‘populations’ by name, e.g, El Azul, should also give the population number (1-10) each time so that it will be easier for the reader to follow what is going on.  Already, most readers will need to refer way back to the geographic map which is early in the manuscript, so adding numbers to names (or names to numbers) for the populations will be useful.

23)  Line 614.  In Figure 2, there seem to be a few very common alleles and some relatively rare alleles that are sampled just once or a few times.  This leads me to suspect that not so much can be said given that some populations were sampled for so few individuals.  I realize that things can still be tested and best interpretations made given limited data, but I think that the skew in sample numbers between ‘populations’ really impacts analyses like these profoundly so the authors should discuss this in a few paragraphs in a detailed way (in particular because so many analyses, so many methods, so many results are presented in this very long paper).  I am left wondering what any of it means when so much is done yet sampling of ‘populations’ and of different marker systems is so skewed.

24)  Line 628.  For Figure 4, I had difficulty seeing what is going on given the poor resolution of the fonts in the figure.

25)  Tables 5, 6, and 7.  Numbers of individuals sampled in each of the populations should be given in each table to give the reader a better feel for differences in sample sizes among populations (which surely has a profound effect on determining significance of differences between populations, or not).

26)  Line 715.  Given the small sample sizes of ‘populations’, I am suspicious of this ‘first generation immigrants’ inference here.  Can such a thing really be inferred given the small samples of individuals in the current study?

27) Line 874.  How does the divergence time between Inia and Pontoporia estimated here compare to prior estimates based on genome-scale clock analyses (and other clock analyses of cetaceans)?  If the estimate here is much shallower or deeper, is this an issue, since an assumption of evolutionary rate (a point estimate?) was made versus prior studies that used many more loci and many fossil calibrations?

28) Figure 15.  How are the reticulations in the network interpreted by the authors considering that mtDNA is generally thought to be maternally inherited and non-recombining?

29) Line 881.  Is this the interpretation if neutrality is assumed and a strict molecular clock is assumed, or are there other interpretations that are also valid, especially if there is mutational heterogeneity and/or natural selection impacts?  Also for the interpretations in paragraph starting on line 888, if selection is in play or errors/uncertainty in the molecular clock are probable, does this change the range of interpretations radically?  Also for results shown in Figure 16.

30) Discussion section (general).  I do fear that the sampling plan of individuals from the different populations, assumptions of clocklike evolution (that match exactly some other species), differences in evolutionary rates between nuclear and mtDNA sequences, and selection may have profound impacts that are not fully discussed in the long Discussion section. 

Comments on the Quality of English Language

The English is generally fine.

Author Response

Referee 3

Referee 3 wrote:

The manuscript is very thorough and detailed and many analyses and results are presented.  I think my main concern is about the sampling of individuals that is very meager for some collection localities relative to others, and the dataset may be overanalyzed given its small size (by modern standards, wherein genome-scale data can be quickly/cheaply collected and analyzed).  The authors might consider the following in revising/improving their manuscript.

Answer:

We thanked and appreciated so much all the comments and observations from Referee 3, which enhances the quality of the manuscript. Now we explained in the text the extreme difficulty to capture the 82 Bolivian pink river dolphins and that effectively some “populations” are composed by few samples. However, we also explained that this question did not have many repercussions for the objectives that we undertake. Two paragraphs are now in the text regarding this question:

(we named them “populations” for operative purposes, but they are really sampling sites. However, these sampling sites belonged to very precise geographical areas, to different rivers, or were separated by different rapids. In fact, it is not very important to define “a priori” operating populations because later certain analyses (such as phylogenetic trees, AMOVA or BAPS will determine exactly how many different populations there are)

This task (to capture the specimens) was hard, difficult, and dangerous because the Bolivian rivers were infected of piranhas, caimans, and anacondas and the indigenous fishermen and the biologists were forced to live in the unpopulated Bolivian jungle for three months. All this did not allow us to obtain the same number of samples from each of the populations analyzed, since, in some places, there were more bufeos and it was, depending on the conditions of the river, easier to capture them. However, in other places, the bufeos were scarcer and the conditions of the river were much more complex to capture them. When it was possible, we employed unbiased statistics for samples sizes.

In a study as the current one, no experimental design for sampling is possible because it is impossible to “a priori” determine where and how many specimens could be caught and the time and resources are very limited inside the Bolivian jungle. Many captures were opportunistic if the conditions of a river were adequate. Contrarily, in other rivers (or trams of the same river), the conditions were extremely negative, and captures were impossible.

Referee 3 said:

1-Line 2.  The title is among the longer ones I have seen for a journal article.  Most simply, could delete “(Control Region, Microsatellites, and DQB-1 Gene Sequences)”?  This level of detail is not necessary as is in the abstract.

Answer:

The title and the Abstract were shortened as suggested Referee 3.

2)   Lines 64-132.  This paragraph is super-long and has a lots of details that may not be necessary.  Cutting some of the very specific details might improve readability since readers can go to the papers cited if want these additional details?  I am thinking maybe reducing length of this paragraph by 50% would better balance the other paragraphs in the introduction.  The study is focused on population structure and genetics, while this long paragraph is related to population threats to the species and some of this could be deleted or moved to the discussion if it ends up being relevant.

Answer:

A considerable fraction of this paragraph has been eliminated following the suggestion of Referee 3.

3)   Line 170.  I would contend that this paragraph has way too much detail and does not really relate to any materials and methods – could cut much of the details here as I do not see how they relate to materials and methods and would increase readability to trim this back.

Answer:

This paragraph has been considerably shortened following the suggestion of Referee 3.

4)   Line 182.  Where the authors say, “54 bufeos were sampled in four different areas (we named them “populations”)”, it might be better to call these ‘sampling sites’ as ‘population’ has a specific (although in my opinion vague…) definition, according to most textbooks.  The sampled sites might represent populations, but at the onset of the study, this is not known, so a more neutral term might be better?

Answer:

The question of the aforementioned “populations” is now explained in the text. Some paragraphs above, it was explained this concept.

5)   Line 200.  In some cases, it seems that very few specimens were sampled for particular ‘populations’.  Is this sampling adequate to determine evolutionary patterns in a robust way?  It seems that many alleles will be missed given this limited sampling of individuals?  However, I realize that sampling more specimens of this rare species may not be easy to achieve.

Answer:

As we explained now in the manuscript, it is very hard and complex to capture river dolphins in the Bolivian Amazon rivers within the Bolivian jungle. In fact, no other specific and detailed study with river dolphins has been carried out by the difficulty to caught specimens of this species in these rivers.

6)   Line 209.  Maybe state at the beginning of this paragraph that valid/approved animal protocols were used in the sampling that comply with Bolivian rules/regulations if such rules/regulations are relevant?

Answer:

Now, we have introduced in the text a paragraph with the legal permissions to capture dolphins and to transport samples of them from Bolivia and Peru to Colombia. The text in the manuscript is now as follows:

The capture, transporting, and ethical permissions to obtain samples of the bufeos were given by the Ministerio de Desarrollo Sostenible y Planificación, Dirección General de Biodiversidad from Bolivia (DGB/UVS No 477/03; approval date May 27 2003) and CITES Bolivia (B09118259), and Ministerio de Producción (Lima, Peru) (No 402-2003-PRODUCE/DNEPP; approval date November 13 2003).

7)   Line 210.  I do not know the latest ‘rules’, but is it appropriate to describe the scientific helpers as ‘Indian fisherman’ or would the fisherman (or perhaps many readers be offended by that wording?  Maybe say ‘native Bolivian fisherman’, ‘indigenous Bolivian fisherman’, or some other?

Answer:

We have changed the expression “Indian fisherman” by “indigenous Bolivian fisherman” following to Referee 3.

8)   Line 221.  For the mt region, it would be helpful to indicate approximately how big the amplicons are to give the reader a feel for how many basepairs of the control region were sequenced?

Answer:

We now explain that the fragment of CR is 400 base pairs.

9)   Line 253.  It is not clear from the methods to this point why the sampling was so different for the various molecular markers (e.g., only 23 individuals were sampled for the nuclear gene, many more individuals sampled for mtDNA).

Answer:

The samples of the bufeos were obtained in 2003. The DNA was extracted several years later. Some samples were small and the quantity and quality of DNA isolated was lower than in other larger samples. The first molecular markers analyzed were mt CR (in fact, we sequenced the complete mitogenome of 81 out 82 dolphins captured-16,400 base pairs). Thus, the DNA of some individuals was over. Later, we analyzed the 10 microsatellites, and the DNA of more specimens was over. Finally, with the DNA of the 23 specimens that have still DNA, we carried out the sequences at DQB-1 gene. For this reason, the sample sizes for the different markers employed is different.

10)  Line 295 and following paragraph.  As noted above, are there enough individuals sampled per ‘population’ to have enough power to do all of these various analyses.  Some of the ‘populations’ included very few individuals, and the a priori grouping into ‘populations’ might be considered somewhat arbitrary (that is, another researcher might have divided sampling localities into more ‘populations’ or merged ‘populations’ into a total lesser number of ‘populations’ from the start)?  More explanation in the methods on how decided on the various ‘populations’ might clarify to the reader exactly why specimens were divided into ‘populations’ the way they were by the authors?

Answer:

Such as we previously explained the sampling sites were operatively named “populations” only by an operative question.  Really, some of the procedures used defined us how many different genetic populations of bufeos are in the Bolivian rivers analyzed.

11)   Line 312.  When did ‘global’ analyses, is it an issue that there is extensive missing data for some samples (e.g., the nuclear gene sampled from few individuals)?  Do the analyses/statistics used correct for/account for the missing data in an adequate way, or is this problematic?  Presumably the simulations deal with the missing data in terms of accounting for the missing data, but a concern is that with such small sample sizes, in particular for the nuclear locus (but also for the other markers), determining whether ‘populations’ are differentiated may be challenging given such small sample sizes?  If this is an issue (i.e., small sample sizes for various populations) that might impact interpretations, the authors should at the least comment on this issue in some way to put the reader at ease?

Answer:

As we explained before the different sample sizes for molecular markers is due to the DNA of several specimens was spent. The fact that in some “populations” the number of specimens was low is correlated with the difficulty of the sampling process. Nevertheless, despite of the heterogeneous number of samples per location, all the markers employed revealed that the genetic structure in the Bolivian rivers analyzed is very limited (small genetic structure). Complete mitogenomes and RAPDs (not shown in this study) reveals the same. Thus, we consider that, despite the low number of samples in some locations, this last event had not a decisive influence in the results obtained. Contrarily, if within the Bolivian rivers there were a lot of many different gene pools, the low samples sizes obtained in some areas should have some repercussion in determining the correct number of different gene pools or Mus. But this was not the case. Likely, we used some unbiased statistics for sample sizes (when it was possible) for the estimate of genetic diversity statistics (and comparisons among them) and in the genetic distances.

12)  Line 360.  These analyses do not assume a priori ‘populations’ which is good, but is it still a concern that sampling of individuals at some collection sites is so sparse?  Or, do many published studies sample so few specimens per collection site?  Again, the randomizations will ‘work’ to account for the low sampling at many sites, but I fear what patterns might be missed by sampling so few individuals per site.  Many analyses were done in this manuscript, but a concern is that a small sample of data has been ‘over-analyzed’ given the amount of data collected.

Answer:

Many published studies with species very difficult to sample have small samples sizes such as we showed here. For instance, for the Wright’s F and genetic heterogeneity statistics as well as for HW equilibrium tests for the microsatellites, we have used randomizations. On the other hand, for example, the spatial pattern analyses were carried with the individuals and not with populations (test de Mantel, spatial autocorrelation, etc). Therefore, for these analyses only is important the total number of individuals studied and not the number of individuals studied by “population”. We consider that the sample sizes we used is perfectly adequate for the analysis we have undertaken and there is not “over-analyzed” data.

13)   Line 397.  Given the importance of this calibration method for the ‘molecular clock’ used (a particular evolutionary rate that is assumed), the authors should note here which taxa were used as a basis for the Inia molecular clock.  A reference is given, but the authors should note here whether the evolutionary rate for the D-Loop is based on this same region being used here and whether the rate is based on cetaceans (which generally have a low rate of molecular evolution among mammals) or some other taxa.

Answer:

We employed the “molecular clock” and the evolutionary substitution rates taken from other papers with Cetaceans. In fact, we used the results of Gravena et al. (2014, 2015), which were obtained for Inia.

14) Line 410.  Is this evolutionary rate assumed here the same or different from the one noted above?

Answer:

Yes, they are the same.

15)  Line 425 and line 437.  Does the ‘Coalescent Constant Population’ model’ assume neutrality and complete absence of selective sweeps?  If not, what is assumed?  If so, is it good to assume there are no selective sweeps, or can selective sweeps be differentiated from population bottlenecks, etc.?  Maybe a bit more here about assumptions of the methods is warranted, although some stuff on selection is mentioned here.

Answer:

Yes, we assume that these models are basically neutral, and we have good evidence for Inia and other mammal species. In fact, we commented in the text how selection or different mutation rates can influence in these results (both comments in Material and methods as well as in the Discussion):

Nevertheless, all these demographic procedures have several caveats. Selection can affect effective population size, reducing the effective number for a time and increasing the coalescence rate later [119]. The same occurs with small changes in the mutation rate (μ), which can greatly affect the effective number and, in turn, estimated divergence time [120].

We know that selection can affect effective population size, reducing the effective number for a time and increasing the coalescence rate later, which could increase “artificially” the temporal splits among populations. The same occurs with small changes in the mutation rate (μ), which can greatly affect the effective number and, in turn, the estimated divergence time. For instance, if the estimated mutation rate is smaller than the real one, the temporal divergence split will be also smaller than the real one for a determined effective size. Contrarily, if the estimated mutation rate is higher than the real one, this increases the temporal divergence estimates in an irreal way.

16)  Line 438.  In addition to changes in mutation rate, the mutation rate assumed, if completely wrong, will make it so that any correlation of changes at the molecular level with biogeographic or bioclimatics (etc.) factors at certain geologic times will potentially be way off.  So, just assuming one particular mutation rate is perhaps not the way to go.  Surely the uncertainty on this molecular rate of control region evolution in Inia is very high, and it would be good to incorporate this broad uncertainty into all analyses/interpretations that center on a molecular clock.  If a strict molecular clock is assumed, the data should be tested to see if the data fit a strict molecular clock or not.

Answer:

This observation made by referee 3 is very certain. However, our main estimate for the process of mitochondrial haplotype diversification for the bufeos in Bolivia is extremely like that obtained by Gravena et al. (2014) for Inia although they used different procedures:

Despite of this, our temporal mtDNA diversification process (180.000-130.000 YA) agrees quite well with that detected by other work [10]. They estimated isolation between the upstream and downstream groups of bufeos across the rapids of the Madeira River to have occurred around 122,000 YA (95 % HPD 32,000-283,000 YA). They performed an IMA2 analysis using a harmonic mean of the average mammalian substitution rates. When they used the upper and lower bound substitution rate values from other study [171], their estimated values ranged from a mean of 97,000 (95 % HPD 25,000–226,000 YA) to 163,000 YA (95 % HPD 43,000-379,000 YA). Henceforth, both studies concluded that at the beginning of the fourth great Pleistocene glaciation, the rapids of the Mamoré-Iténez (Guaporé) River basin and the Madeira River basin influenced the diversification of mitochondrial haplotypes in I. boliviensis.

17)  Line 505.  I am not personally familiar with the wide variety of microsatellite analyses done, so I leave comments on these analyses to other reviewers who have experience running microsatellites and analyzing them.

Answer:

Ok, there is not problem.

18)  Line 522.  I did not understand the basis for the statement, “These mitochondrial genetic diversity levels are medium for the CR.”  There should be some sort of citation(s) here to explain what ‘medium’ is.  Is this ‘medium’ for a cetacean, for a mammal, for a vertebrate, for vertebrates and invertebrates?

Answer:

It is medium compared to a wide number of other species of mammals.

19) Line 548.  For the nuclear locus, a concern may be that the limited number of specimens analyzed did not allow a rigorous test of whether deviations from HW equilibrium is the case?

Answer:

We use for this task the exact probabilities of the G test, with the Markov chain method and with a 10,000-dememorization number, 20 batches and 5,000 iterations per batch, which is the most powerful test for small samples as the present one.

20)  Line 561.  The divergence times of these different genera of ‘river dolphins’ have been estimated many times and shown not to be particularly closely related, so this is either a point that is not worth making or can say that this result is consistent with basically all previous molecular clock analyses of cetaceans that have been done properly (e.g., various McGowen et al. papers).

Answer:

Effectively, we explained in the text exactly that commented by Referee 3:

It is interesting to note that Pontoporia blainvillei is the sister species of Inia and the Asian River dolphin (Platanista gangetica) is not closely related with Inia for CR. This result is consistent with basically all previous molecular clock analyses of cetaceans that have been done properly [132].

21) Line 579.  In this table, I think it would be useful to show explicitly how many individuals were sampled for each of the 10 ‘populations’ shown since some ‘populations’ had many individuals and others had very few.  This would help readers interpret the results better maybe?

Answer:

We included in all the tables the sample sizes (n) of each one of the “populations” employed, such as suggested Referee 3.

22)  Line 599 and other areas.  When refer to ‘populations’ by name, e.g, El Azul, should also give the population number (1-10) each time so that it will be easier for the reader to follow what is going on.  Already, most readers will need to refer way back to the geographic map which is early in the manuscript, so adding numbers to names (or names to numbers) for the populations will be useful.

Answer:

According to Referee 3, we included together the name of the populations also the population number for more facility for the readers.

23)  Line 614.  In Figure 2, there seem to be a few very common alleles and some relatively rare alleles that are sampled just once or a few times.  This leads me to suspect that not so much can be said given that some populations were sampled for so few individuals.  I realize that things can still be tested and best interpretations made given limited data, but I think that the skew in sample numbers between ‘populations’ really impacts analyses like these profoundly so the authors should discuss this in a few paragraphs in a detailed way (in particular because so many analyses, so many methods, so many results are presented in this very long paper).  I am left wondering what any of it means when so much is done yet sampling of ‘populations’ and of different marker systems is so skewed.

Answer:

This is discussed, for instance, in some part of the Discussion (this is also applicable to question 28):

Additionally, the reticulations observed in the MJN agrees well with this scenario. Some lineages of bufeos should be isolated in several riverine refuges when the levels of water decreased during the last phase of the Pleistocene and accumulated unique mutations, but when the waters rose again some lineages expanded and widely dispersed their genetic characteristics across the major fraction of the Mamoré River basin. For this reason, we found exclusive and private haplotypes together with well-expanded haplotypes in a same population.

24)  Line 628.  For Figure 4, I had difficulty seeing what is going on given the poor resolution of the fonts in the figure.

Answer:

We tried to obtain a better resolution with this Figure 4 and others.

25)  Tables 5, 6, and 7.  Numbers of individuals sampled in each of the populations should be given in each table to give the reader a better feel for differences in sample sizes among populations (which surely has a profound effect on determining significance of differences between populations, or not).

Answer:

We added the sample sizes for all the populations studied in all the tables.

26)  Line 715.  Given the small sample sizes of ‘populations’, I am suspicious of this ‘first generation immigrants’ inference here.  Can such a thing really be inferred given the small samples of individuals in the current study?

Answer:

We believe that the assignation analysis is useful even when the sample sizes of some “populations” were small. Precisely, if different gene pools were present due to fragmentation induced by the rapids, small groups of dolphins in each side of the rapids, showed marked differences associated with founder effects and it should be very difficult to determine migrants of first generation. However, the assignation analysis showed frequent cases of first migration generation, with many cases with individuals of very distant areas. This puts forward that the rapids (main objective of our work) have a minimal influence in the genetic structure of the Bolivian dolphins despite the small sample sizes for some “populations”.

27) Line 874.  How does the divergence time between Inia and Pontoporia estimated here compare to prior estimates based on genome-scale clock analyses (and other clock analyses of cetaceans)?  If the estimate here is much shallower or deeper, is this an issue, since an assumption of evolutionary rate (a point estimate?) was made versus prior studies that used many more loci and many fossil calibrations?

Answer:

We have now compared our divergence times between Inia and Pontoporia with other works:

With the MJN procedure, and using the CR data (Figure 15), the split between Inia and Pontoporia was dated between 15.121 + 0.154 to 13.170 + 0.138 MYA. This temporal divergence between both genera (15-13 MYA) is highly like that obtained in other studies. From a paleontological point of view, at least, a minimal temporal split between the ancestors of Inia and Pontoporia was estimated to be around 13-11 MYA [139]. From a molecular perspective, three different studies with different procedures yielded between 17.6-8.8 MYA [3], 13 MYA [140], and 17-15 MYA [132]. Thus, the time split obtained between both genera was very similar among all these different studies.

28) Figure 15.  How are the reticulations in the network interpreted by the authors considering that mtDNA is generally thought to be maternally inherited and non-recombining ?

Answer:

It was answered in question 23.

29) Line 881.  Is this the interpretation if neutrality is assumed and a strict molecular clock is assumed, or are there other interpretations that are also valid, especially if there is mutational heterogeneity and/or natural selection impacts?  Also for the interpretations in paragraph starting on line 888, if selection is in play or errors/uncertainty in the molecular clock are probable, does this change the range of interpretations radically?  Also for results shown in Figure 16.

Answer:

It was commented in the Discussion:

We know that selection can affect effective population size, reducing the effective number for a time and increasing the coalescence rate later, which could increase “artificially” the temporal splits among populations. The same occurs with small changes in the mutation rate (μ), which can greatly affect the effective number and, in turn, the estimated divergence time. For instance, if the estimated mutation rate is smaller than the real one, the temporal divergence split will be also smaller than the real one for a determined effective size. Contrarily, if the estimated mutation rate is higher than the real one, this increases the temporal divergence estimates in an irreal way.

30) Discussion section (general).  I do fear that the sampling plan of individuals from the different populations, assumptions of clocklike evolution (that match exactly some other species), differences in evolutionary rates between nuclear and mtDNA sequences, and selection may have profound impacts that are not fully discussed in the long Discussion section.

Answer:

We explicitly commented in the final of the Discussion:

The use of different kinds of molecular markers is relevant as we showed in the current work because some of them are considered to have a neutral behavior (mt CR which is illustrative of the female colonization process, and nu microsatellites, which detected the important vector of gene flow which the males are), but others (DQB-1 locus) clearly respond to pathogen selective pressure which are similar in different Neotropical rivers.

For all these reasons, for instance, the correlograms from the spatial autocorrelation analysis are different for each one of these different molecular markers.

Comments on the Quality of English Language

The English is generally fine.

Answer:

Thanks so much.

Thanks so much for all your comments and suggestions, which greatly enhanced the quality of the manuscript.

Prof. Manuel Ruiz-García. PhD